# ESCAPING LOW-RANK TRAPS: INTERPRETABLE VISUAL CONCEPT LEARNING VIA IMPLICIT VECTOR QUANTIZATION

**Shujian Gao**[1,2]*, **Yuan Wang**[3], **Chenglong Ma**[1,2], **Xin Gao**[1,2], **Jiangtao Yan**[2], **Junzhi Ning**[2],

**Cheng Tang**[2], **Changkai Ji**[1,2], **Huihui Xu**[2], **Wei Li**[2], **Ziyan Huang**[2], **Jiashi Lin**[2], **Ming Hu**[2],

**Jiyao Liu**[2], **Wenhao Tang**[2], **Ye Du**[2], **Tianbin Li**[2], **Jin Ye**[2], **Junjun He**[2]†

[1]Fudan University    [2]Shanghai AI Laboratory    [3]Zhejiang University

## ABSTRACT

Concept Bottleneck Models (CBMs) achieve interpretability by interposing a human-understandable concept layer between perception and label prediction. We first identify that the condition of *many-to-many* mapping is necessary for robust CBMs, a prerequisite that has been largely overlooked in previous approaches. While several recent methods have attempted to establish this relationship, we observe that they suffer from the fundamental issue of *representation collapse*, where visual patch features degenerate into a low-rank subspace during training, severely degrading the quality of learned concept activation vectors, thus hindering both model interpretability and downstream performance. To address these issues, we propose Implicit Vector Quantization (IVQ), a lightweight regularizer that maintains high-rank, diverse representations throughout training. Rather than imposing a hard bottleneck via direct quantization, IVQ learns a codebook prior that anchors semantic information in visual features, allowing it to act as a proxy objective. To further exploit these high-rank concept-aware features, we propose Magnet Attention, which dynamically aggregates patch-level features into visual concept prototypes, explicitly modeling the many-to-many vision–concept correspondence. Extensive experimental results show that our approach effectively prevents representational collapse and achieves state-of-the-art performance on diverse benchmarks. Our experiments further probe the low-rank phenomenon in representational collapse, finding that IVQ mitigates the information bottleneck and yields cross-modal representations with clearer, more interpretable consistency. Code is available at https://github.com/Daryl-GSJ/IVQ-CBM.

## 1 INTRODUCTION

Explainable Artificial Intelligence (xAI) aims to embed neural networks with *human–interpretable* and *interactive* reasoning processes, thereby opening the *black box* of end-to-end prediction systems. Among the ante-hoc xAI methods, the **Concept Bottleneck Model** (CBM) (Koh et al., 2020) is a prominent approach that pipelines predictions through an intermediate *concept layer*. This layer, situated between a perceptual encoder and a final task head, forces the model to operate in two distinct stages: First, a *perception* stage maps inputs to a set of predefined semantic concepts (*e.g.*, *shape of a beak* or *spatial extent of a lesion*). Second, a *reasoning* stage uses only these concept activation vectors (CAVs) to make the final decision.

In the two-stage learning, perception and modeling of CAVs in the initial stage is foundational. The core of this stage lies in constructing a cross-modal patch-concept alignment process, which compels

---

*This work was completed during my internship at Shanghai AI Lab.
†Corresponding author.

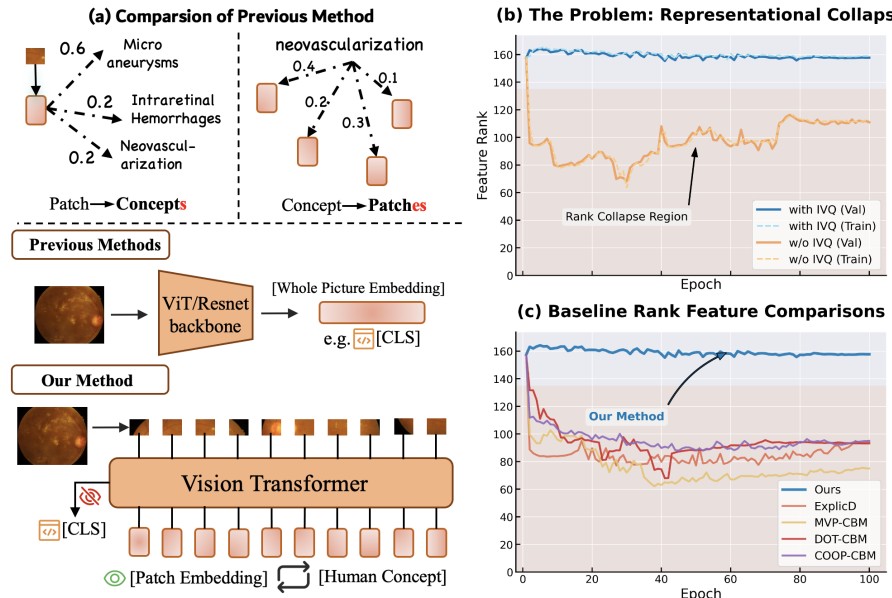

Figure 1: (a) An illustration of many-to-many cross-modal alignment in CBMs. (b) Comparison of feature rank dynamics during training on the training and validation sets, with and without IVQ. (c) Feature rank dynamic comparisons with previous baselines.

the model to learn the mapping and disentanglement of raw, high-dimensional visual embeddings into a set of structured visual representation vectors corresponding to human-defined concepts. As illustrated in Figure 1a, an intrinsic many-to-many correspondence exists between local visual features and high level semantic concepts within the cross-modal concept alignment. The relationship is two-fold: an individual image patch may map to multiple concepts, while concurrently, the visual representation of a single concept is distributed across several distinct image patches. However, previous methods such as (Yang et al., 2023; Yuksekgonul et al., 2023; Oikarinen et al., 2023; Sheth & Kahou, 2023) employ a visual encoder to extract a single global embedding for concept alignment. Due to the limited expressiveness of a single visual feature, these approaches fail to explicitly model the complex many-to-many mapping between local features and high-level concepts.

However, when we attempt to explicitly construct this relationship, a critical phenomenon termed *representational collapse* emerges, severely impeding the process. As presented in Figure 1b, we tracked the rank of the patch feature matrix throughout the training; the rank undergoes a sharp decline during the initial epochs on both training and validation sets, eventually bottoming out at a rank of 70, a drastic reduction from the potential full rank of 196. This behavior is not an anomaly; as shown in Figure 1c, concurrent works such as ExplicD (Gao et al., 2024) and MVP-CBM (Wang et al., 2025) exhibit a similar pattern, suggesting that *representational collapse* is a fundamental obstacle in this domain. Meanwhile, its essence is a symptom of a more fundamental problem: the collapse of feature diversity. This collapse is particularly damaging for CBMs, as a degenerated representation space lacks the expressive capacity to encode a diverse set of concepts (Sansone et al., 2025). When visual feature embeddings become highly similar and informationally redundant, the learning process itself is confounded (Jing et al., 2022), directly inhibiting the formation of high-quality, disentangled visual features, which are the foundation of the CBM.

Hence, we introduce Implicit Vector Quantization (IVQ), a novel regularizer that repurposes the VQ objective as a loss term without quantizing the forward pass. The loss effectively forms a semantic bridge, compelling each patch feature to align with the nearest learned codebook prototype. Collectively, these prototypes act as distinct anchors that prevent the feature distribution from collapsing into a degenerate subspace. As depicted in Figure 1b and c, this directly counters representation collapse by maintaining a stable and elevated feature rank throughout training. Furthermore, building upon concept-aware, well-structured visual features, we introduce a Magnet Attention mechanism, which dynamically aggregates the diverse patch-level features into a holistic, semantically coher-

ent visual representation prototype for each pre-defined textual concept, effectively modeling the many-to-many mapping.

This work makes three key contributions:

- We identify that the key to CBMs lies in modeling the many-to-many relationship between concepts and patches. Furthermore, we identify and analyze *representational collapse*, a key challenge in training modern CBMs that hinders the establishment of CAVs.

- We propose IVQ, a novel regularization method that preserves feature diversity and prevents representational collapse without creating an information bottleneck. To exploit obtained rich representations, we introduce the Magnet Attention mechanism to effectively aggregate the regularized patch features into semantically meaningful concept prototypes.

- Extensive experiments on **eight benchmarks** demonstrate that our IVQ-CBM consistently outperforms **eight strong baselines**, achieving state-of-the-art accuracy and learning better interpretable representations consistent with textual concepts.

## 2 RELATED WORK

**Concept Bottleneck Model.** It achieves interpretability-by-design by introducing a human-defined concept layer that bridges raw visual features and human-understandable semantics, providing a foundational explanation for the model's final decision (Koh et al., 2020). A key challenge lies in achieving high-quality cross-modal alignment, *i.e.*, constructing a precise, fine-grained mapping between visual features and textual concepts. However, many popular methods (Yuksekgonul et al., 2023; Oikarinen et al., 2023) rely on a single, holistic visual feature, such as the `[CLS]` token from CLIP-style models or a global image embedding from visual foundation models (Kim et al., 2023).

These approaches operate on the premise that a global feature vector encapsulates all necessary visual attributes (Zhang et al., 2014; Raghu et al., 2022). This assumption is untenable in complex visual scenes, especially for medical images characterized by subtle, intricate, and fragmented lesions (Chen et al., 2021).

To establish a more fine-grained, many-to-many mapping, recent works have begun to leverage patch-level embeddings from Vision Transformers (ViTs) (Dosovitskiy et al., 2021), which offer a powerful prior for perception and alignment. Several works have ventured in this direction, employing techniques such as Optimal Transport (Xie et al., 2025), or Cross-Attention and dynamic pooling to learn visual concepts (Wang et al., 2025; Gao et al., 2024), yet a critical issue arises: the visual features extracted from the encoder suffer from **representational collapse**, a phenomenon where the feature vectors degenerate into a low-dimensional subspace, becoming informationally redundant and lacking diversity.

**Representation Regularization.** Low-rank issue has been extensively studied in the self-supervised learning (SSL) literature. For instance, Barlow Twins (Zbontar et al., 2021) mitigates collapse by minimizing the redundancy between feature dimensions via a cross-correlation matrix. Other techniques, such as spectral regularization (Yoshida & Miyato, 2017), constrain the spectral norm of weight matrices to improve generalization. A more recent work DINOv3 (Siméoni et al., 2025) introduces gram regularization loss to prevent the model's output from collapsing to a trivial solution.

Nevertheless, a critical drawback exists in directly applying these techniques to CBMs: they are not tailored for the specific demands of the CBM task, nor are they designed for cross-modal, many-to-many alignment. Specifically, the core objective of these regularization methods is to *indiscriminately* maximize feature diversity or reduce redundancy (Huang et al., 2017; Gao & Pu, 2025). For CBMs, however, the goal is not arbitrary diversity, but rather **meaningful, structured diversity** that aligns with human-defined concepts(Gao et al., 2026a). Indiscriminate decorrelation or spectral regularization cannot guarantee that the learned feature diversity has any correspondence with the predefined semantic concepts. Consequently, these methods may amplify trivial visual details that are useless or even detrimental to the final task, thereby interfering with the formation of high-quality concept vectors (as we demonstrate in Section 4.2).

## 3 METHODOLOGY

### 3.1 PRELIMINARIES AND OVERALL FRAMEWORK

**Problem Formulation.** Consider a dataset of triplets $\mathcal{D} = \{(x_i, c_i, y_i)\}_{i=1}^N$, where $x_i \in \mathcal{X}$ is an input image, $y_i \in \mathcal{Y}$ denotes the corresponding class label, and $c_i$ is a set of textual descriptions defining the concepts associated with class $y_i$. In contrast to standard black-box models that directly learn a mapping $x_i \rightarrow y_i$, the CBM pipeline is formulated as a two-stage process: $x_i \rightarrow c_i \rightarrow y_i$. First, the alignment stage produces a CAV, which we denote as $\mathbf{v}_i \in \mathbb{R}^K$. This vector contains the activation scores for all $K$ concepts for a given image $x_i$. It is generated by aligning the learned Visual Concepts, $\mathcal{M} \in \mathbb{R}^{K \times D}$, with their corresponding textual concept embeddings, $\boldsymbol{\tau} \in \mathbb{R}^{K \times D}$. Specifically, the activation score for the $k$-th concept is computed as the dot product between its visual and textual representations:

$$(\mathbf{v}_i)_k = \mathcal{M}_k \cdot \boldsymbol{\tau}_k, \quad \forall k \in \{1, \dots, K\}. \tag{1}$$

The CAVs then serve as the exclusive input for the second-stage classification task. Given the straightforward mechanism of the second stage, achieving precise and efficient image-concept alignment in the first stage is of paramount importance.

**Visual Concept Learning.** To acquire fine-grained visual features, we leverage a pre-trained CLIP ViT encoder to obtain a structured visual representation from the input image $x_i$:

$$\mathbf{Z}_v = [\mathbf{z}_{\text{cls}}, \mathbf{z}_1, \mathbf{z}_2, \dots, \mathbf{z}_L] = E_I(x_i) \in \mathbb{R}^{(L+1) \times D}. \tag{2}$$

where $\mathbf{z}_{\text{cls}}$ is the class token embedding and $\mathbf{Z}_p = [\mathbf{z}_1, \dots, \mathbf{z}_L] \in \mathbb{R}^{L \times D}$ represents the matrix of patch token embeddings. This patch-level representation is crucial, as a single textual concept often corresponds to information spanning multiple patches, while a single patch may contain details relevant to multiple concepts.

This complex, many-to-many mapping necessitates a sophisticated mechanism to bridge the gap between local features and high-level concepts. Therefore, our objective is to aggregate the patch features $\mathbf{Z}_p$ into $K$ meaningful **Visual Concepts**, denoted by the matrix $\mathcal{M} \in \mathbb{R}^{K \times D}$. These learned Visual Concepts are subsequently used for the final concept-text alignment as described in our problem formulation. The following sections detail the two core components of our method designed to achieve this: IVQ for representation regularization and the Magnet Attention mechanism for concept aggregation.

### 3.2 IMPLICIT VECTOR QUANTIZATION

To mitigate the aforementioned issues, we propose a novel IVQ mechanism. In contrast to conventional VQ approaches, which first identify the nearest codebook vector over a distance metric and subsequently propagate the resulting quantized features through the forward pass, our IVQ method strategically discards the quantized features. Instead, it exclusively leverages the commitment and codebook losses as regularization terms during backpropagation. This process compels each visual patch embedding to align more closely with its nearest codebook vector—which semantically corresponds to a textual concept, as will be verified in our Section 4.2. This, in turn, distills the core conceptual information while simultaneously regularizing the representation space and enhancing the diversity of visual features, ultimately benefiting the subsequent aggregation of visual concepts.

Given the encoded visual patch features $\mathbf{Z}_p = \{\mathbf{z}_j\}_{j=1}^L$, a critical challenge is to prevent representation collapse and enhance feature diversity. We maintain a small, learnable codebook $\mathcal{C}_{vq} \in \mathbb{R}^{M \times D}$, where M equals the number of textual concepts K (discussed in Section 4.2). For each patch feature $\mathbf{z}_j \in \mathbf{Z}_p$, we find the nearest codebook vector $\mathbf{c}_k \in \mathcal{C}_{vq}$ via an `argmin` operation over the Euclidean distance:

$$k_j = \underset{k}{\text{argmin}} \|\mathbf{z}_j - \mathbf{c}_k\|_2^2. \tag{3}$$

The quantized representation $\mathbf{z}_{q,j} = \mathbf{c}_{k_j}$. However, we discard this quantized output $\mathbf{Z}_q$ and do not use it in the subsequent forward pass. Instead, we compute the VQ loss, which consists of a codebook loss and a commitment loss, to update the feature encoder and $\mathcal{C}_{vq}$:

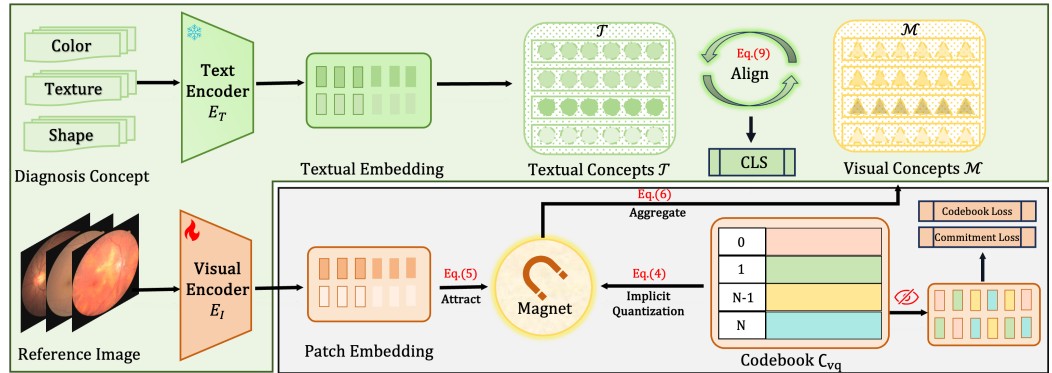

Figure 2: A pipeline of the proposed IVQ-CBM method which optimizes training from three dimensions: classification accuracy, concept alignment, as well as representation diversity and quality. An many-to-many mapping between visual embeddings and textual concepts establish the foundation of our model. IVQ further regularizes the representation space and helps to maintain and distill the core information for each patch throughout training. Building upon obtained high-rank concept-aware features, magnet aggregates visual concepts from raw embeddings, regarded as input for concept alignment and final classification.

$$\mathcal{L}_{\text{IVQ}} = \underbrace{\|\text{sg}(\mathbf{Z}_p) - \mathbf{Z}_q\|_2^2}_{\text{Codebook Loss}} + \beta \underbrace{\|\mathbf{Z}_p - \text{sg}(\mathbf{Z}_q)\|_2^2}_{\text{Commitment Loss}}, \quad (4)$$

where $\text{sg}(\cdot)$ denotes the stop-gradient operator and $\beta$ is the commitment cost hyperparameter. By backpropagating $\mathcal{L}_{\text{IVQ}}$, we compel the patch features in $\mathbf{Z}_v$ to align with a learned prototype without suffering from the information bottleneck of hard quantization. This process regularizes the representation space, encouraging a more structured and diverse feature distribution, which is crucial for the subsequent aggregation stage.

### 3.3 MAGNET CONCEPT AGGREGATION

With a regularized, high-rank feature space established by IVQ, we now introduce a mechanism to bridge the gap between low-level visual features and high-level semantic concepts. A simple spatial pooling of all patch features would lose critical fine-grained information (Wang et al., 2025). Therefore, we propose the **Magnet Attention** mechanism, a differentiable soft-clustering module designed to aggregate the $L$ patch features into $K$ semantically meaningful Visual Concepts.

To achieve this, we first introduce a set of learnable **concept queries**, denoted as $\mathbf{Q} \in \mathbb{R}^{K \times D}$, where $K$ is the number of concepts. Each query vector $\mathbf{q}_k \in \mathbf{Q}$ acts as a learnable center-point to attract patch features related to a specific concept. For the input patch features $\mathbf{Z}_p = \{\mathbf{z}_j\}_{j=1}^L$, we compute a similarity score between each patch feature $\mathbf{z}_j$ and each concept query $\mathbf{q}_k$. Following common practice, we use negative squared Euclidean distance as the similarity metric (Van Den Oord et al., 2017b).

This similarity is then converted into a soft assignment matrix $\mathbf{A} \in \mathbb{R}^{L \times K}$ via a softmax function over the concepts:

$$A_{jk} = \frac{\exp(-\|\mathbf{z}_j - \mathbf{q}_k\|_2^2)}{\sum_{k'=1}^K \exp(-\|\mathbf{z}_j - \mathbf{q}_{k'}\|_2^2)}, \quad (5)$$

where $A_{jk}$ represents the soft-assignment weight of the $j$-th patch to the $k$-th concept query. These weights form an attention map over the patches for each concept. The final **Visual Concepts**, denoted by the matrix $\mathcal{M} \in \mathbb{R}^{K \times D}$, are then computed as a weighted average of the patch features:

$$\mathcal{M} = \mathbf{A}^\top \mathbf{Z}_p. \quad (6)$$

The resulting matrix $\mathcal{M}$ contains $K$ rich visual concept prototypes, each summarizing the relevant spatial information from the image corresponding to a distinct semantic concept. This aggregated representation $\mathcal{M}$ is then used for the final alignment with the textual concept embeddings $\tau$.

### 3.4 TRAINING OBJECTIVES

We train our model end-to-end with a composite loss that simultaneously optimizes for task accuracy, concept alignment, and representation quality.

**Classification Loss.** The primary objective is to ensure the model accurately performs the final classification task. Let $h_{\mathrm{cls}}$ denote the final classification head. It takes the vector of concept activation scores $\mathbf{v}_i \in \mathbb{R}^K$ as input to produce the final class logits $\mathbf{p}_i = h_{\mathrm{cls}}(\mathbf{v}_i)$. As defined previously, each score $(\mathbf{v}_i)_k$ represents the alignment between the $k$-th visual concept and the $k$-th textual concept. The classification loss $\mathcal{L}_{\mathrm{cls}}$ is the standard cross-entropy between the predicted logits and the ground-truth class label $y_i$:

$$\mathcal{L}_{\mathrm{cls}} = \mathcal{L}_{\mathrm{CE}}(\mathbf{p}_i, y_i). \tag{7}$$

**Concept Alignment Loss.** To ensure the model's interpretability, we explicitly supervise the concept bottleneck. The concept activation scores $\mathbf{v}_i \in \mathbb{R}^K$ serve as the logits for concept prediction. We supervise these logits against the ground-truth concept labels $\mathbf{c}_i \in \{0,1\}^K$, which is a multi-hot vector indicating the presence of each of the $K$ concepts for the sample $x_i$. The concept alignment loss $\mathcal{L}_{\mathrm{concept}}$ is the binary cross-entropy (BCE) loss between the predicted concept scores and the ground-truth labels:

$$\mathcal{L}_{\mathrm{concept}} = \mathcal{L}_{\mathrm{BCE}}(\mathbf{v}_i, \mathbf{c}_i). \tag{8}$$

This loss is crucial as it forces the model to learn a set of visually grounded and semantically meaningful concepts.

**IVQ Regularization Loss.** As detailed previously, the $\mathcal{L}_{\mathrm{IVQ}}$ encourages the patch-level features to form a more structured and diverse representation space without introducing a hard information bottleneck. This improves the robustness of the feature extractor and aids the Magnet Attention mechanism in forming higher-quality Visual Concepts.

**Overall Training Objective.** The final training objective combines these three components into a single multi-task loss function. The total loss $\mathcal{L}$ is formulated as a weighted sum:

$$\mathcal{L} = \mathcal{L}_{\mathrm{cls}} + \mathcal{L}_{\mathrm{concept}} + \mathcal{L}_{\mathrm{IVQ}}. \tag{9}$$

## 4 EXPERIMENTS AND DISCUSSIONS

**Core Questions.** We structure our analysis around a series of research questions (RQs) to thoroughly investigate the properties and efficacy of our proposed method. Our goal is to dissect its underlying mechanisms, justify its design choices, and demonstrate its advantages in terms of performance and interpretability.

**RQ1: Impact on Representation Collapse.** How does implicit quantization affect the rank of the learned representations? Figure. 4 and 6.

**RQ2: Efficacy of Implicit Quantization.** Is implicit quantization a genuinely effective technique? How does it conceptually and empirically differ from explicit quantization? Section. 4.2

**RQ3: Implicit Quantization as a Regularizer.** Our method can be interpreted as a form of representation space regularization. How does its performance compare against other established regularization techniques? Why does our proposed implicit quantization, which leverages a codebook, outperform traditional regularization techniques on CBM? Section. 4.2

**RQ4: Optimal Codebook Configuration.** The core component of our method is the codebook, which is dynamically updated throughout training. What is the optimal relationship between the codebook size, $M$, and the number of textual concept vectors, $K$? Section. 4.2

**RQ5: Interpretability as a Visual Dictionary.** Beyond performance, how does the codebook contribute to model interpretability? Section. 4.2

**Baselines, Benchmarks, and Metrics.** To comprehensively evaluate the robustness and generalization of our method, we conduct experiments on a diverse suite of public benchmarks spanning two distinct domains. First, for **medical imaging**, we utilize datasets covering dermoscopy (Codella

et al., 2019), histopathology (Kather et al., 2018), fundus photography (Porwal et al., 2018; yiwe-ichen04, 2021), ultrasound (Al-Dhabyani et al., 2020), chest X-ray (Zawacki et al., 2019; Johnson et al., 2019), and mammography (Cui et al., 2021). Second, to demonstrate the broad **generalizability** of IVQ-CBM to natural images, we extend our evaluation to five standard computer vision benchmarks: CIFAR-10 and CIFAR-100 (Krizhevsky, 2009) for general object classification, CUB-200-2011 (Wah et al., 2011) for fine-grained identification, Places365 (Zhou et al., 2017) for scene recognition, and ImageNet-1K (Russakovsky et al., 2015) for large-scale classification.

To demonstrate its effectiveness, we benchmark our approach against eight recent and popular CBM methods: LaBo (Yang et al., 2023), PCBM (Yuksekgonul et al., 2023), COOP-CBM (Sheth & Kahou, 2023), LF-CBM (Oikarinen et al., 2023), Explicd (Gao et al., 2024), MVP-CBM (Wang et al., 2025), CLEAR (Dikter et al., 2024), and DOT-CBM (Xie et al., 2025). Besides, following (Gao et al., 2024; Wang et al., 2025), we also compare black-box models (He et al., 2015; Dosovitskiy et al., 2021) and multi-modal backbones (Radford et al., 2021; Wang et al., 2022; Zhang et al., 2025).

To account for the class imbalance prevalent in these benchmarks, we employ both Accuracy (ACC) and Balanced Macro Average Accuracy (BMAC) for a comprehensive evaluation. For implementation details, please refer to Appendix D.

Table 1: Overall Performance comparison (ACC %) on multiple medical and general benchmark datasets. We evaluate our model against various zero-shot, black-box, and explainable models. Best results are highlighted in **bold**. Gains ($\Delta$) are compared within the Explainable methods group.

| Model | Medical Datasets | | | | | | | | General Benchmarks | | | | |
|---|---|---|---|---|---|---|---|---|---|---|---|---|---|
| | ISIC | NCT | IDRID | BUSI | Retina | SIIM | Cardio | CMMD | C-10 | C-100 | CUB | Plc365 | ImgNet |
| *Zero-shot* | | | | | | | | | | | | | |
| CLIP [ICML 2021] | 29.88 | 26.67 | 29.84 | 43.85 | 18.33 | 41.01 | 54.81 | 27.10 | - | - | - | - | - |
| MedCLIP [EMNLP 2022] | 13.68 | 9.57 | 16.86 | 27.44 | 17.50 | 42.34 | 45.94 | 27.10 | - | - | - | - | - |
| BiomedCLIP [NEJM AI 2025] | 59.83 | 40.51 | 39.15 | 29.36 | 53.33 | 39.88 | 49.13 | 52.90 | - | - | - | - | - |
| *Black-box* | | | | | | | | | | | | | |
| ResNet50 [CVPR 2016] | 83.53 | 92.12 | 54.38 | 76.41 | 78.33 | 87.97 | 81.51 | 74.59 | - | - | - | - | - |
| ViT Base [ICLR 2021] | 90.01 | 93.25 | 58.07 | 80.25 | 83.33 | 83.56 | 79.60 | 74.77 | - | - | - | - | - |
| *Explainable* | | | | | | | | | | | | | |
| LaBo [CVPR 2023] | 79.20 | 91.73 | 50.77 | 84.01 | 72.60 | 74.13 | 73.48 | 70.78 | 80.23 | 60.17 | 69.88 | 39.67 | 68.04 |
| PCBM [ICLR 2023] | 85.91 | 91.77 | 54.36 | 84.64 | 66.67 | 80.02 | 76.43 | 69.24 | 84.61 | 63.22 | 72.36 | 41.13 | 70.14 |
| COOP-CBM [Nips 2023] | 86.82 | 93.43 | 61.22 | 89.61 | 85.00 | 86.14 | 80.57 | 77.28 | 85.17 | 64.21 | 73.06 | 42.19 | 71.23 |
| LF-CBM [ICLR 2023] | 83.55 | 87.92 | 60.59 | 76.77 | 73.33 | 77.62 | 75.95 | 74.75 | 82.94 | 61.89 | 71.42 | 40.33 | 69.46 |
| Explicd [MICCAI 2024] | 88.72 | 95.29 | 63.26 | 87.17 | 83.33 | 85.70 | 78.95 | 76.74 | 86.03 | 64.91 | 74.08 | 43.07 | 71.93 |
| MVP-CBM [IJCAI 2025] | 87.72 | 97.90 | 65.38 | 89.74 | 85.00 | 84.89 | 80.29 | 75.84 | 86.72 | 65.48 | 74.63 | 43.81 | 72.29 |
| CLEAR [WACV 2025] | 86.25 | 89.05 | 56.54 | 83.85 | 72.89 | 78.26 | 78.22 | 73.35 | 83.77 | 62.83 | 71.84 | 40.92 | 68.81 |
| DOT-CBM [CVPR 2025] | 86.55 | 90.15 | 58.45 | 85.23 | 74.59 | 79.52 | 77.15 | 71.15 | 84.38 | 63.45 | 72.29 | 41.76 | 69.31 |
| **IVQ-CBM (Ours)** | **90.11** | **99.90** | **67.35** | **93.59** | **88.33** | **88.03** | **82.01** | **79.25** | **88.14** | **67.12** | **75.91** | **45.54** | **73.42** |
| +$\Delta$ | +1.39 | +2.00 | +1.97 | +3.85 | +3.33 | +1.89 | +1.44 | +1.97 | +1.42 | +1.64 | +1.28 | +1.73 | +1.13 |

## 4.1 BASIC EXPERIMENTS

**Benchmark Comparison.** As demonstrated in our experiments in Table 1 and Table 2, our method consistently outperforms all baselines across the full range of medical datasets, highlighting its robust performance under diverse diagnostic conditions. Notably, our approach excels over traditional black-box models, maintaining high diagnostic accuracy while providing interpretability. This is a significant contribution to resolving the long-standing trade-off between performance and interpretability in prior works (Zarlenga et al., 2022).

To further validate the scalability and generalization of our framework, we extended our evaluation to widely used general-domain benchmarks. As shown in Table 1 and Table 2, IVQ-CBM demonstrates superior performance across all five datasets, spanning from fine-grained classification tasks such as CUB to large-scale challenges like ImageNet. Noteworthy improvements are observed over the strongest baselines, including +1.64% on CIFAR-100 and +1.13% on ImageNet, confirming that our method's benefits scale effectively to complex, high-dimensional visual distributions.

Additionally, to probe the underlying mechanisms behind these performance improvements, we analyzed the feature rank dynamics during training. Figure 4 and Figure 5 present the rank evolution across medical and general benchmarks, respectively. The results reveal a fundamental distinc-

Table 2: Overall Performance comparison (BMAC %) on multiple medical and general benchmark datasets. We evaluate our model against various zero-shot, black-box, and explainable models. Best results are highlighted in **bold**. Gains ($\Delta$) are compared within the Explainable methods group.

| Model | Medical Datasets | | | | | | | | General Benchmarks | | | | |
|---|---|---|---|---|---|---|---|---|---|---|---|---|---|
| | ISIC | NCT | IDRID | BUSI | Retina | SIIM | Cardio | CMMD | C-10 | C-100 | CUB | Plc365 | ImgNet |
| *Zero-shot* | | | | | | | | | | | | | |
| CLIP [ICML 2021] | 21.32 | 26.71 | 25.92 | 37.85 | 25.00 | 22.15 | 49.12 | 34.00 | - | - | - | - | - |
| MedCLIP [EMNLP 2022] | 15.31 | 12.05 | 19.57 | 43.19 | 25.42 | 41.01 | 46.66 | 30.72 | - | - | - | - | - |
| BiomedCLIP [NEJM AI 2025] | 24.47 | 40.39 | 30.31 | 36.45 | 38.33 | 20.92 | 46.70 | 50.00 | - | - | - | - | - |
| *Black-box* | | | | | | | | | | | | | |
| ResNet50 [CVPR 2016] | 76.53 | 91.34 | 55.88 | 75.84 | 77.50 | 76.21 | 81.51 | 60.46 | - | - | - | - | - |
| ViT Base [ICLR 2021] | 84.14 | 92.73 | 54.70 | 83.63 | 76.66 | 74.15 | 79.61 | 60.35 | - | - | - | - | - |
| *Explainable* | | | | | | | | | | | | | |
| LaBo [CVPR 2023] | 80.83 | 91.62 | 54.17 | 85.98 | 73.83 | 72.16 | 73.73 | 65.59 | 79.15 | 59.95 | 69.72 | 39.41 | 67.88 |
| PCBM [ICLR 2023] | 81.76 | 91.18 | 58.85 | 88.91 | 69.17 | 75.68 | 76.41 | 66.68 | 84.49 | 63.07 | 72.19 | 40.99 | 69.97 |
| COOP-CBM [Nips 2023] | 79.25 | 93.78 | 50.41 | 89.73 | 80.83 | 79.81 | 80.58 | 64.25 | 84.99 | 64.03 | 72.87 | 42.01 | 71.09 |
| LF-CBM [ICLR 2023] | 78.46 | 87.77 | 56.97 | 75.77 | 70.83 | 71.59 | 75.95 | 60.18 | 82.78 | 61.73 | 71.22 | 40.15 | 69.21 |
| Explicd [MICCAI 2024] | 82.42 | 94.73 | 63.61 | 87.37 | 81.67 | 78.90 | 78.94 | 60.48 | 85.88 | 64.78 | 73.91 | 42.89 | 71.77 |
| MVP-CBM [IJCAI 2025] | 80.35 | 97.89 | 57.78 | 91.45 | 83.33 | 78.95 | 80.29 | 56.15 | 86.54 | 65.30 | 74.45 | 43.66 | 72.15 |
| CLEAR [WACV 2025] | 81.11 | 86.86 | 58.33 | 83.33 | 70.56 | 70.12 | 78.18 | 61.23 | 83.59 | 62.61 | 71.69 | 40.75 | 68.66 |
| DOT-CBM [CVPR 2025] | 81.37 | 91.51 | 59.12 | 84.10 | 71.88 | 76.48 | 77.10 | 62.98 | 84.19 | 63.28 | 72.11 | 41.53 | 69.17 |
| **IVQ-CBM (Ours)** | **86.22** | **99.88** | **73.06** | **95.38** | **85.83** | **81.91** | **82.01** | **69.70** | **87.91** | **66.88** | **75.68** | **45.32** | **73.23** |
| +$\Delta$ | +3.80 | +1.99 | +9.45 | +3.93 | +2.50 | +2.10 | +1.43 | +3.02 | +1.37 | +1.58 | +1.23 | +1.66 | +1.08 |

tion between our method and prior approaches. While competing methods, particularly DOT-CBM (Xie et al., 2025) and MVP-CBM, exhibit varying levels of representational collapse, our method consistently maintains a high and stable feature rank across all datasets. Notably, the collapse is exacerbated on more complex datasets, such as ImageNet, as shown in Figure 5, where the feature ranks of baseline models dramatically decline. This provides compelling evidence that IVQ acts as a strong regularizer, effectively preventing the deterioration of the feature space. By preserving high-rank and diverse representations, this structural advantage directly correlates with the superior and more stable downstream performance observed in Table 2.

**Ablation Study.** Our ablation study (Table 3) confirms that both the IVQ and Magnet modules are integral to performance. IVQ provides the most significant boost, especially on the BMAC metric for imbalanced datasets like IDRID (+11.81). Removing the Magnet module and reverting to a standard [CLS] token baseline leads to a notable performance drop. This indicates that relying on a single global feature vector is an oversimplification of the image's content. While this approach may suffice for coarse-grained classification, it is inadequate for complex scenes requiring fine-grained analysis, such as localizing small or scattered targets—a common challenge in medical imaging (Chen et al., 2021). Furthermore, we find that IVQ's mechanism is linked to preventing *representation collapse*(Figure 6). Models without IVQ suffer from rank collapse (dashed lines), whereas our method maintains a high, stable feature rank (solid lines). This shows IVQ acts as a powerful regularizer, ensuring diverse and robust feature learning.

## 4.2 EXTENSIVE EXPERIMENTS

**Vector Quantization versus Implicit Vector Quantization.** As previously established, an inherent many-to-many correspondence exists between visual patches and textual concepts, with each patch often encoding multi-faceted semantic information. Standard VQ conflicts with this principle (Van Den Oord et al., 2017b). Its use of a hard `argmin` operation maps each patch to a single, nearest codebook vector, which forces the collapse of a patch's rich information into a discrete representation. Consequently, only these quantized features are passed forward to the magnet module, discarding other relevant visual attributes and violating the many-to-many relationship, leading to the critical **information bottleneck** issue. In contrast, our IVQ is designed to resolve this issue. It retains the quantization objective solely as a regularizer, encouraging each raw patch feature to align with its nearest prototype. Crucially, this allows the original, high-fidelity feature vector to be used in the forward pass for the magnet module. This process distills the core conceptual information from the patch, enabling a more effective concept alignment. Results are summarized in Table 4, revealing that IVQ substantially outperforms the explicit quantization across all six datasets, on both Classification and Concept BMAC metrics.

Table 3: Ablation experiments showing both ACC and BMAC metrics for each experimental setup. The final rows display the performance gains (+Δ in red) of our full model over the second-best configuration.

| Components | | Metric | Datasets | | | | | | | |
|---|---|---|---|---|---|---|---|---|---|---|
| IVQ | Magnet | | ISIC | NCT | IDRID | BUSI | Retina | SIIM | Cardio | CMMD |
| ✗ | ✔ | ACC | 80.88 | 95.23 | 57.14 | 88.46 | 75.00 | 85.72 | 80.21 | 74.59 |
| | | BMAC | 84.84 | 94.92 | 45.27 | 87.37 | 69.16 | 78.45 | 80.21 | 52.55 |
| ✔ | ✗ | ACC | 89.42 | 96.23 | 65.38 | 92.31 | 83.33 | 86.47 | 81.11 | 77.63 |
| | | BMAC | 82.85 | 95.61 | 61.25 | 91.15 | 80.00 | 80.49 | 81.11 | 66.81 |
| ✔ | ✔ | ACC | **90.11** (+Δ 0.69) | **99.90** (+Δ 3.67) | **67.35** (+Δ 1.97) | **93.59** (+Δ 1.28) | **88.33** (+Δ 5.00) | **88.03** (+Δ 1.56) | **82.01** (+Δ 0.90) | **79.25** (+Δ 1.62) |
| | | BMAC | **86.22** (+Δ 1.38) | **99.88** (+Δ 4.27) | **73.06** (+Δ 11.81) | **95.38** (+Δ 4.23) | **85.83** (+Δ 5.83) | **81.91** (+Δ 1.42) | **82.01** (+Δ 0.90) | **69.70** (+Δ 2.89) |

Table 4: **Ablation study of different representation quantization methods.** We compare the performance on various medical datasets in terms of Classification BMAC and Concept BMAC. The improvement of our **Implicit Quantization (IVQ)** over the Explicit baseline is highlighted in parentheses (+Δ). Our method consistently outperforms the baseline, particularly in preventing representation collapse.

| Dataset | Explicit Quantization | | Implicit Quantization (Ours) | |
|---|---|---|---|---|
| | Cls. BMAC | Con. BMAC | Cls. BMAC | Con. BMAC |
| ISIC | 51.47 | 22.22 | **86.22** (+34.75) | **83.88** (+61.66) |
| Cardio | 60.61 | 53.30 | **82.01** (+21.40) | **80.25** (+26.95) |
| BUSI | 57.28 | 37.48 | **95.38** (+38.10) | **71.02** (+33.54) |
| SIIM | 50.00 | 50.00 | **81.91** (+31.91) | **80.15** (+30.15) |
| CMMD | 50.35 | 50.00 | **69.70** (+19.35) | **66.66** (+16.66) |
| IDRID | 23.75 | 50.85 | **73.06** (+49.31) | **59.48** (+8.63) |

Table 5: Performance comparison of our model and other representation regularization techniques across datasets. Gains (+Δ) and losses (-Δ) are shown relative to the best-performing alternative technique.

| Technique | SIIM | | ISIC | | BUSI | |
|---|---|---|---|---|---|---|
| | ACC | BMAC | ACC | BMAC | ACC | BMAC |
| Barlow Twins (Zbontar et al., 2021) | 86.39 | 79.17 | 87.62 | 81.16 | 91.02 | 88.81 |
| Spectral Regularization (Yoshida & Miyato, 2017) | 85.64 | 80.84 | **90.21** | 84.74 | 92.30 | 91.98 |
| Gram Loss (Siméoni et al., 2025) | 84.56 | 74.07 | 76.64 | 59.83 | 84.61 | 82.68 |
| **IVQ (Ours)** | **88.03** (+1.64) | **81.91** (+1.07) | 90.11 (-0.10) | **86.22** (+1.48) | **93.59** (+1.29) | **95.38** (+3.40) |

**Representation Regularization Techniques.** Several representation regularization techniques have been proposed to mitigate feature rank collapse, particularly in SSL. We benchmark our proposed IVQ against three prominent methods: explicit de-correlation via Barlow Twins (Zbontar et al., 2021), Spectral regularization (Yoshida & Miyato, 2017), and Gram Loss (Siméoni et al., 2025). As shown in Table 5, IVQ consistently outperforms these general-purpose regularization techniques across most metrics, yielding significant gains over the strongest baseline.

A critical question raises: is a higher feature rank directly correlated with superior CBM performance? An analysis of feature rank dynamics, presented in Figure 7, reveals a more nuanced relationship. While methods like Barlow Twins and Spectral regularization often maintain a higher feature rank than IVQ, particularly on the ISIC and BUSI datasets, this elevated rank does not translate to better downstream performance, suggesting a form of **over-regularization**. We attribute this to a key distinction: unlike general-purpose methods that indiscriminately maximize feature diversity, IVQ is tailored to foster the **meaningful, structured diversity** required by CBMs by aligning features with a set of learnable prototypes.

**Analysis of Codebook Size** The IVQ codebook dynamically aligns patch features with a set of learnable prototypes, which ensures feature diversity while distilling core visual information and

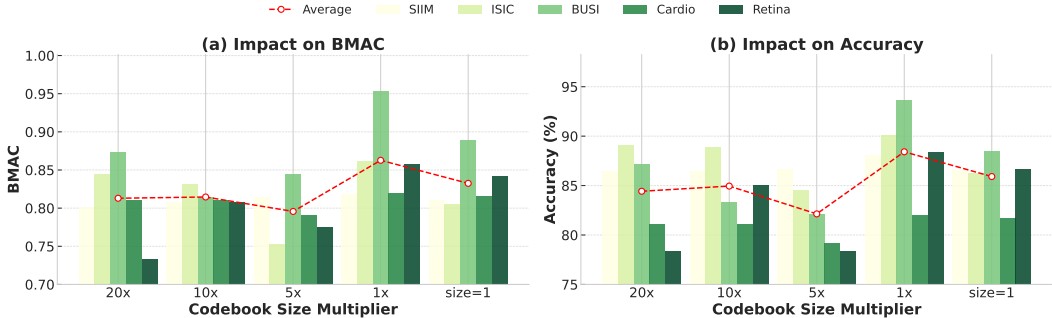

Figure 3: Performance (ACC & BMAC) across various codebook sizes. The size is defined as a multiple ($\times$) of the number of textual concepts, $K$.

avoiding the bottleneck of conventional VQ. This raises a key question regarding the optimal size of the codebook, $M$, in relation to the number of textual concepts, $K$. To investigate this, we conduct an ablation study, setting the codebook size to various multiples of the concept count (i.e., $M = \alpha K$ for $\alpha \in \{20, 10, 5, 1\}$), as well as a baseline with a single shared prototype ($M = 1$).

As illustrated in Figure 3, the model's performance is robust across a range of oversized codebooks ($20\times$, $10\times$, and $5\times$), but we observe a distinct peak in both average ACC and BMAC when the codebook size is set equal to the number of textual concepts ($M = K$). This setting achieves the highest average BMAC and Accuracy across all datasets. We hypothesize that setting $M = K$ encourages a **one-to-one mapping** between the learned visual prototypes and the predefined textual concepts, fostering a more structured and semantically aligned representation space. It provides strong evidence for the critical role of our IVQ as a bridge between visual patches and textual concepts, acting as a system of visual semantic anchors. These results suggest that aligning the codebook's capacity with the task's conceptual granularity is the optimal strategy.

**Visualization of CodeBook in IVQ**   The IVQ continuously distills visual information for each concept during training. To qualitatively evaluate the knowledge captured within the final codebook, we visualize the index maps of our learned prototypes on a representative mammogram from the CMMD dataset, as shown in Figure 8a.

The resulting mapping demonstrates a highly logical process that mirrors clinical reasoning. The prototype for `Mass Margin` (light green) precisely delineates the lesion's contour, while those for `Mass Shape` (cyan) and `Calcification Features` (blue) correspond to its internal characteristics. Crucially, the prototype for `Associated Features` (dark blue) extends its focus beyond the lesion's border, probing the surrounding parenchyma for signs of structural distortion— a key indicator of malignancy. This structured, multi-faceted assessment validates that our codebook has learned semantically meaningful and clinically relevant concept representations (For a more detailed visual analysis, please refer to the appendix F).

## 5   CONCLUSION

In this work, we propose **IVQ-CBM**, which explicitly models the many-to-many relationship while addressing representation collapse. Our method features two key components. IVQ, which uses a learnable codebook prior to anchor visual patches, and Magnet Attention, which aggregates these patches into semantically coherent visual concepts aligned with textual definitions. Extensive experiments demonstrate that IVQ-CBM achieves superior performance over baselines without sacrificing interpretability, with ablation studies validating each component's contribution. Our analysis reveals that IVQ circumvents the hard information bottleneck of direct VQ and, unlike general regularization methods, fosters a *meaningful* feature diversity that is more effective than indiscriminately maximizing a mathematical objective. Visualizations confirm that our approach yields high-quality, interpretable visual concept representations that are consistent with their textual concepts, resulting in a more faithful and robust form of interpretability for CBMs.

## REPRODUCIBILITY STATEMENT

To ensure reproducibility, we release the source code in the supplementary materials. All datasets used in our experiments are either publicly available; the implementation details, including model architectures, hyper-parameters, and optimization settings, are described in section 4 and Appendix D. For baselines, we rely on publicly released implementations and adapt them with the same preprocessing pipeline as described in section 4.1. We hope these materials enable the community to faithfully reproduce our results and extend our approach.

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

# A    APPENDIX

In this section, we present additional implementation details, experiment results, theoretical analysis, pseudo code, and supplements. The content structure is outlined as follows:

- Section B - Remaining Related Work
- Section C - Theoretical Analysis
    - Section C.1 - Motivation: The Fundamental Limitations of a Single Global Feature
    - Section C.2 - Proof of Intrinsic Many-to-Many Coupling
    - Section C.3 - Theoretical Analysis: Optimization Dynamics and Rank Preservation
    - Section C.4 - Theoretical Analysis for Gradient Convergence
- Section D - Implementation Details
- Section E - Feature Rank Dynamics
- Section F - Visualization of Codebook in IVQ
- Section G - Visualization of Codebook in VQ
- Section H - Visualization Analysis With Baselines
- Section I - Results with Other Multi-Modal Backbone
- Section J - Statement on the Use of Large Language Models
- Section K - Pseudocode of IVQ-CBM
- Section L - Quantitative Interpretability and Casual Faithfulness
- Section M - Analysis of Information Leakage and Effective Concept Learning
- Section N - Sensitivity Analysis of Commitment Cost Parameter

# B    REMAINING RELATED WORK

**Vector Quantization.**  VQ has played a central role in learning discrete latent representations. Early work on Neural Discrete Representation Learning (Van Den Oord et al., 2017a) introduced codebook-based quantization that enables end-to-end training through a nearest-neighbor commitment objective. Follow-up variants such as VQ-VAE-2 (Razavi et al., 2019) further enhanced generative fidelity via multi-level discrete hierarchies. Beyond architectural extensions, a series of studies examined the optimization challenges of VQ. Rotation-based VQ (Fifty et al., 2025) restructures the quantization space to stabilize code assignments, whereas linear-layer VQ (Zhu et al., 2025) proposes lightweight transformations to mitigate representation collapse and improve codebook utilization. VQ has also been explored within broader perceptual tasks: Vector-Quantized Vision Foundation Models (Zhao et al., 2025) leveraged discrete visual tokens for object-centric learning, and scaling studies, such as 100k-VQGAN (Zhu et al., 2024), demonstrated that extremely large codebooks can maintain high utilization. More recent designs like MGVQ (Jia et al., 2025) adopt multi-group quantization to increase representational granularity and improve generalization.

Despite these advances, prior works rely on explicit quantization, where the quantized vectors replace continuous features in the forward pass. Such a hard discrete bottleneck is often beneficial for generative modeling but is misaligned with CBMs, which require preserving rich, distributed semantic information and allowing many-to-many relations between patches and concepts.

# C    THEORETICAL ANALYSIS

## C.1    MOTIVATION: THE FUNDAMENTAL LIMITATIONS OF A SINGLE GLOBAL FEATURE

A prevailing and influential paradigm in the evolution of Concept Bottleneck Models is the use of a single global feature from a pre-trained vision model, such as CLIP, as the sole representation of an input image. Specifically, these models typically rely on the final [CLS] token from a Vision Transformer (ViT) or the feature vector generated by a Global Average Pooling (GAP) layer in a Convolutional Neural Network (CNN) (He et al., 2015). The implicit assumption underpinning

this approach is that a single high-dimensional vector (e.g., $D = 768$) can sufficiently and comprehensively encapsulate all semantic information relevant to downstream tasks. In other words, it posits that this single, condensed vector is a complete basis for identifying all pertinent concepts and, ultimately, the final class label.

However, we argue that this seemingly efficient simplification is, in fact, an *illusion of sufficiency* that conceals fundamental flaws. While it may suffice for simple, object-centric classification tasks, its limitations are starkly exposed when confronted with visually complex scenes that demand fine-grained understanding.

Our central thesis is that the reliance on a single global feature vector is an **untenable design simplification** for building robust, interpretable, and broadly applicable CBMs. This simplification is particularly fragile in scenarios where:

- The image contains multiple, spatially distinct objects or concepts.
- The critical differentiating information lies in subtle, local details or textures.
- The key features are inherently distributed or non-centralized.

This argument is not speculative but is grounded in the unavoidable theoretical limitations inherent to the mechanisms that generate this global vector. We prove this by deconstructing its three fatal flaws.

**1. Irreversible Loss of Spatial Information.** The generation of a single global vector is, by its very nature, a process that destroys spatial structure. For a CNN-based GAP layer, the mechanism averages a feature map of size $H \times W \times D$ across its spatial dimensions to produce a $1 \times 1 \times D$ vector. This operation is mathematically *permutation-invariant*; as long as the set of local features remains the same, their spatial arrangement can be arbitrarily altered with little to no change in the resulting global vector. This is analogous to calculating the word frequency of a document while discarding the sentences and paragraphs. We know *what* content is present, but we have permanently lost *how* it is organized. Consequently, any task requiring answers to "where?" or an understanding of spatial relationships is impossible for a model that has discarded all coordinate information.

For the Transformer-based `[CLS]` token, while its computation involves a spatially-aware self-attention mechanism, the process is ultimately one of *aggregation and summarization*. The final `[CLS]` output is an abstract vector that has encoded spatial relationships into its dimensions, but the explicit, original topological structure is lost (Park & Kim, 2022). The attention maps exist during computation, but the final vector itself does not retain this map. **Conclusion:** A single global vector actively discards the two-dimensional structure of an image, which is its first fatal flaw (Yu et al., 2022).

**2. The Information Bottleneck and Feature Suppression.** A vector of fixed dimensionality has a finite capacity for information, creating a natural bottleneck (Butakov et al., 2024; Gao et al., 2026b). Within an image, different regions, objects, and textures must compete for representation within this fixed bandwidth. In this process of *feature competition*, strong signals (from large, prominent objects) will disproportionately dominate the final vector's representation, while weak signals (from small or subtle objects) are easily averaged out or *suppressed* (Li et al., 2023). Consider a thought experiment: a high-resolution CT scan where 99.9% of the image consists of healthy lung tissue (a strong signal) and only 0.1% contains a small, early-stage pulmonary nodule (a weak signal). During the global aggregation process, the features representing healthy tissue will overwhelm the vector, making it nearly impossible for the faint but critical signal from the nodule to survive this democratic aggregation. **Conclusion:** The limited capacity of a single global vector forces a lossy compression that systematically sacrifices the fine-grained or low-prevalence information critical for complex tasks.

**3. Representational Failure for Multiple Instances and Concepts.** The core promise of a CBM is a clear mapping from image features to concepts. A single global vector is powerless to manage this when "many-to-many" relationships are required. When multiple independent semantic concepts coexist in an image (e.g., "striped texture," "pointed ears," and "furry texture"), a single vector is forced to *entangle* these disparate pieces of information (Li et al., 2024). This entanglement

leads to severe *representational ambiguity*. An activation in a 21 dimension might represent concept A, concept B, or an inseparable combination of A and C. This fundamentally violates the goal of CBMs to be interpretable and intervenable. An ideal concept bottleneck should have distinct "channels" corresponding to disentangled concepts (Xie et al., 2025). For example, an intervention on the "pointed ears" concept becomes meaningless if its representation is coupled with the "furry texture" concept, as we cannot modify one without affecting the other. **Conclusion:** To model multiple concepts independently and clearly, a model requires access to multiple, separable visual features prior to the concept alignment stage. A single global vector cannot meet this fundamental requirement.

In summary, drawing from the current paradigm of CBMs that rely on a single global feature, we have proven through theoretical analysis that this is a fundamentally flawed simplification. Its three fatal flaws—the loss of spatial information, the creation of an information bottleneck, and the failure to represent multiple concepts clearly—are inherent to its generation mechanism. This provides a solid and compelling theoretical foundation for our proposed paradigm: establishing a many-to-many mapping between local image features (patches) and semantic concepts.

### C.2 PROOF OF INTRINSIC MANY-TO-MANY COUPLING

**Proposition 1.** *Within the proposed IVQ-CBM framework, (i) each learned Visual Concept inevitably depends on multiple visual patch embeddings, and (ii) each visual patch embedding inevitably contributes to multiple Visual Concepts.*

*Proof.* We prove the two parts of the proposition separately.

### 1. FORMAL DEFINITIONS AND NOTATION

We adopt the notation from the method section.

**Visual Patch Embeddings** $\mathbf{Z}_p = [\mathbf{z}_1, \ldots, \mathbf{z}_L] \in \mathbb{R}^{L \times D}$ are the patch embeddings from the ViT encoder $\mathcal{E}_v$.

**Concept Queries** $\mathbf{Q} = [\mathbf{q}_1, \ldots, \mathbf{q}_K]^\top \in \mathbb{R}^{K \times D}$ is the matrix of learnable concept queries.

**Soft Assignment Matrix** $\mathbf{A} \in \mathbb{R}^{L \times K}$, where its element $A_{jk}$ represents the soft-assignment weight of patch $j$ to concept $k$:

$$A_{jk} = \frac{\exp(-\|\mathbf{z}_j - \mathbf{q}_k\|_2^2)}{\sum_{k'=1}^{K} \exp(-\|\mathbf{z}_j - \mathbf{q}_{k'}\|_2^2)}.$$

**Visual Concepts** $\mathbf{M} = [\mathbf{m}_1, \ldots, \mathbf{m}_K]^\top = \mathbf{A}^\top \mathbf{Z}_p \in \mathbb{R}^{K \times D}$ is the matrix of aggregated Visual Concepts. The $k$-th visual concept is given by $\mathbf{m}_k = \sum_{j=1}^{L} A_{jk} \mathbf{z}_j$.

**Concept Prediction** The activation score for concept $k$, denoted $v_k$, is the cosine similarity between the visual concept $\mathbf{m}_k$ and its corresponding textual concept embedding $\boldsymbol{\tau}_k$, i.e., $v_k = \frac{\mathbf{m}_k \cdot \boldsymbol{\tau}_k}{\|\mathbf{m}_k\|\|\boldsymbol{\tau}_k\|}$. The final prediction is $\hat{c}_k = \sigma(v_k)$, where $\sigma$ is the sigmoid function for the BCE loss.

### 2. PROOF: "ONE CONCEPT–MULTIPLE PATCHES" IS ALMOST CERTAIN

This proposition states that for any given Visual Concept $\mathbf{m}_k$, its final representation and subsequent prediction $\hat{c}_k$ are influenced by multiple patch embeddings.

**Lemma 1** (Sufficiency). *Let $A_{jk}$ be the soft-assignment weight for patch $j$ and concept $k$. If there exist two distinct patch indices $i \neq j$ such that their assignment weights to concept $k$ are non-zero (i.e., $A_{ik} > 0$ and $A_{jk} > 0$), then the concept prediction $\hat{c}_k$ simultaneously depends on both patch embeddings $\mathbf{z}_i$ and $\mathbf{z}_j$, as demonstrated by their non-zero gradients:*

$$\frac{\partial \hat{c}_k}{\partial \mathbf{z}_i} \neq 0 \quad and \quad \frac{\partial \hat{c}_k}{\partial \mathbf{z}_j} \neq 0.$$

*Proof of Lemma.* We apply the chain rule to compute the gradient of the concept prediction $\hat{c}_k$ with respect to a patch embedding $\mathbf{z}_j$. Since $\frac{\partial \hat{c}_k}{\partial \mathbf{z}_j} = \sigma'(v_k) \frac{\partial v_k}{\partial \mathbf{z}_j}$ and $\sigma'(v_k) \neq 0$ almost everywhere, we only need to analyze the gradient of the activation score, $\frac{\partial v_k}{\partial \mathbf{z}_j}$.

$$\frac{\partial v_k}{\partial \mathbf{z}_j} = \frac{\partial v_k}{\partial \mathbf{m}_k} \frac{\partial \mathbf{m}_k}{\partial \mathbf{z}_j}.$$

The visual concept $\mathbf{m}_k = \sum_{l=1}^{L} A_{lk} \mathbf{z}_l$. The derivative of $\mathbf{m}_k$ with respect to $\mathbf{z}_j$ involves two components, as the assignment weights $A_{lk}$ also depend on $\mathbf{z}_j$:

$$\frac{\partial \mathbf{m}_k}{\partial \mathbf{z}_j} = \underbrace{A_{jk} \cdot \mathbf{I}}_{\text{Direct Term}} + \underbrace{\sum_{l=1}^{L} \frac{\partial A_{lk}}{\partial \mathbf{z}_j} \otimes \mathbf{z}_l}_{\text{Attention Term}},$$

where $\mathbf{I}$ is the identity matrix and $\otimes$ is the outer product. The *Direct Term* $A_{jk} \cdot \mathbf{I}$ is non-zero as long as the assignment weight $A_{jk}$ is non-zero. This term captures the direct contribution of patch $\mathbf{z}_j$ to the weighted average. The *Attention Term* captures the indirect influence of changing $\mathbf{z}_j$ on the attention weights of all other patches and is generally non-zero.

Since $A_{jk}$ is strictly positive (due to the exponential function in softmax), the Direct Term ensures that $\frac{\partial \mathbf{m}_k}{\partial \mathbf{z}_j}$ is non-zero. The gradient $\frac{\partial v_k}{\partial \mathbf{m}_k}$ is also non-zero in general. Therefore, the total gradient $\frac{\partial \hat{c}_k}{\partial \mathbf{z}_j}$ is non-zero. This holds for any patch $j$ with a non-zero assignment weight, proving the lemma. $\square$

**Probability = 1 Argument.** The condition of the lemma, $A_{ik} > 0$ and $A_{jk} > 0$, is certain to hold. The assignment weight $A_{jk}$ is calculated as $\exp(-\|\mathbf{z}_j - \mathbf{q}_k\|_2^2)$ divided by a sum of such terms. Since the squared Euclidean distance is finite and the exponential function is strictly positive for all finite inputs, every assignment weight $A_{jk}$ is strictly greater than zero for any $j \in \{1, \ldots, L\}$ and $k \in \{1, \ldots, K\}$. Consequently, every patch has a non-zero influence on every concept. The "one concept–multiple patches" relationship is not just almost certain; it is a structural certainty of the Magnet Attention mechanism.

## 3. Proof: "One Patch–Multiple Concepts" is Almost Certain

This proposition states that a single patch embedding $\mathbf{z}_j$ will meaningfully contribute to the formation of multiple Visual Concepts.

**Gradient-based Argument.** Consider the update to a patch embedding $\mathbf{z}_j$ during backpropagation, driven by the concept alignment loss $\mathcal{L}_{\text{concept}} = \sum_{k=1}^{K} \mathcal{L}_{\text{BCE}}(\hat{c}_k, c_k)$. The gradient of the loss with respect to $\mathbf{z}_j$ is:

$$\begin{aligned}
\frac{\partial \mathcal{L}_{\text{concept}}}{\partial \mathbf{z}_j} &= \sum_{k=1}^{K} \frac{\partial \mathcal{L}_{\text{concept}}}{\partial \hat{c}_k} \frac{\partial \hat{c}_k}{\partial \mathbf{z}_j} \\
&= \sum_{k=1}^{K} \underbrace{\left( \frac{\hat{c}_k - c_k}{\hat{c}_k (1 - \hat{c}_k)} \right)}_{\text{Error Signal}} \underbrace{\frac{\partial \hat{c}_k}{\partial \mathbf{z}_j}}_{\text{Patch Influence}}
\end{aligned} \quad (10)$$

As established in the previous section, the influence term $\frac{\partial \hat{c}_k}{\partial \mathbf{z}_j}$ is non-zero for all concepts $k$. Therefore, if the model has a non-zero prediction error for more than one concept (i.e., $\hat{c}_k - c_k \neq 0$ for multiple $k$), the total gradient received by patch $\mathbf{z}_j$ will be a sum of contributions from all those concepts. Since natural images typically contain features relevant to multiple concepts, the SGD update will pull $\mathbf{z}_j$ in a direction that is a composite of gradients from multiple concepts, making patch $\mathbf{z}_j$ influential for all of them.

**Softmax Competition Argument.** We can formalize the argument that a single patch is unlikely to contribute to only one concept by quantitatively analyzing the conditions required for the assignment distribution to become nearly one-hot.

Let's start by introducing a temperature parameter $\tau > 0$ into the softmax function, which controls the sharpness of the output distribution. The assignment weight of patch $j$ to concept $k$ is given by:

$$A_{jk} = \frac{\exp(-\|\mathbf{z}_j - \mathbf{q}_k\|_2^2/\tau)}{\sum_{k'=1}^K \exp(-\|\mathbf{z}_j - \mathbf{q}_{k'}\|_2^2/\tau)}.$$

For the original formulation, we can simply consider $\tau = 1$. A "one-hot" assignment, where patch $j$ contributes almost exclusively to a single concept, means that for some concept $k^*$, its weight $A_{jk^*}$ approaches 1.

Let $k^*$ be the index of the concept query closest to the patch embedding $\mathbf{z}_j$:

$$k^* = \arg\min_{k \in \{1,...,K\}} \|\mathbf{z}_j - \mathbf{q}_k\|_2^2.$$

To quantify the significance of this closest distance, we define the **minimum distance margin** $\Delta_{\min}$ for patch $\mathbf{z}_j$ as the difference between the squared distance to the second-closest query and the closest one:

$$\Delta_{\min} \triangleq \min_{k' \neq k^*} \left( \|\mathbf{z}_j - \mathbf{q}_{k'}\|_2^2 - \|\mathbf{z}_j - \mathbf{q}_{k^*}\|_2^2 \right).$$

A large $\Delta_{\min} \geq 0$ indicates that $\mathbf{z}_j$ has a clear "winner" concept query, while a small $\Delta_{\min}$ suggests high competition. Now, we establish a lower bound for the maximum assignment weight $A_{jk^*}$:

$$A_{jk^*} = \frac{1}{1 + \sum_{k' \neq k^*} \exp\left(-\frac{\|\mathbf{z}_j - \mathbf{q}_{k'}\|_2^2 - \|\mathbf{z}_j - \mathbf{q}_{k^*}\|_2^2}{\tau}\right)}.$$

By definition, $\|\mathbf{z}_j - \mathbf{q}_{k'}\|_2^2 - \|\mathbf{z}_j - \mathbf{q}_{k^*}\|_2^2 \geq \Delta_{\min}$ for any $k' \neq k^*$. This allows us to bound the sum in the denominator, yielding the inequality:

$$A_{jk^*} \geq \frac{1}{1 + (K-1)e^{-\Delta_{\min}/\tau}}.$$

This inequality provides a precise condition for a one-hot assignment. For $A_{jk^*}$ to be nearly 1 (e.g., $A_{jk^*} \geq 1 - \epsilon$), the margin-to-temperature ratio $\Delta_{\min}/\tau$ must be large, specifically $\Delta_{\min} \geq \tau \log\left(\frac{K-1}{\epsilon}\right)$. In other words, a patch can only be exclusively assigned to one concept if its embedding $\mathbf{z}_j$ is **geometrically well-separated** from all but one concept query.

Such a large margin is geometrically improbable in high-dimensional spaces due to the **concentration of measure phenomenon** (Vershynin, 2018), which states that distances between random points tend to be tightly clustered. It is far more likely that $\mathbf{z}_j$ will be reasonably close to several queries, leading to a small $\Delta_{\min}$ and thus a distributed (non-sparse) set of assignment weights. The soft nature of the attention mechanism, combined with these geometric properties, ensures that each patch almost certainly contributes to multiple concepts.

## 4. CONCLUSION

The Magnet Attention mechanism, by its design, establishes a dense, many-to-many coupling between patch embeddings and visual concepts.

1. **One Concept to Multiple Patches**: This occurs with structural certainty due to the nature of the soft attention mechanism where all weights are non-zero.

2. **One Patch to Multiple Concepts**: This is a highly probable outcome under standard training conditions. The competitive nature of the softmax function makes a one-hot assignment (where one patch contributes to only one concept) an unstable and non-generic solution in a high-dimensional space. Furthermore, the training dynamics, driven by multi-concept error signals, actively steer patch embeddings to be useful for multiple concepts.

Therefore, the patch-concept mapping in the proposed model is intrinsically many-to-many, providing a robust foundation for learning comprehensive and interpretable visual concepts. $\quad\square$

### C.3 THEORETICAL ANALYSIS: OPTIMIZATION DYNAMICS AND RANK PRESERVATION

In this section, we provide a formal analysis of how the optimization dynamics of the proposed IVQ loss explicitly counteract representational collapse and preserve the rank of the feature space.

**1. Optimization Dynamics as a Restoring Force.** Recall the commitment loss component of the IVQ objective defined in Eq. 4:

$$\mathcal{L}_{commit} = \beta \sum_{j=1}^{L} ||z_j - sg(c_{k_j})||_2^2, \tag{11}$$

where $z_j \in \mathbb{R}^D$ is the $j$-th patch feature, and $c_{k_j}$ is its nearest neighbor in the codebook $\mathcal{C}_{vq}$. During backpropagation, the gradient of this loss with respect to a specific patch feature $z_j$ is given by:

$$\nabla_{z_j} \mathcal{L}_{commit} = 2\beta(z_j - c_{k_j}). \tag{12}$$

This gradient can be interpreted physically as a *restoring force* in the high-dimensional feature space. It actively pulls every patch embedding $z_j$ towards its assigned semantic prototype $c_{k_j}$. Unlike standard contrastive losses that primarily push features apart, this dynamic acts as a structured gravitational pull, clustering the continuous distribution of patch features into compact regions centered around the learned prototypes.

**2. Geometric Interpretation and Rank Lower Bound.** Representational collapse manifests as a rapid decay in the singular values of the feature matrix $Z_p \in \mathbb{R}^{L \times D}$, causing features to degenerate into a low-dimensional subspace (i.e., $\text{rank}(Z_p) \ll K$). The IVQ mechanism imposes a geometric constraint that counteracts this degeneracy.

Empirically, as shown in our ablation study ($M = K$) and UMAP visualizations (Figure 9), the codebook prototypes $\{c_1, \ldots, c_K\}$ converge to a set of well-separated, linearly independent vectors. Geometrically, these prototypes span a support subspace $\mathcal{S}_{code} \subset \mathbb{R}^D$ with an effective rank of approximately $K$.

By minimizing the commitment loss $\mathcal{L}_{IVQ}$, the optimization process acts as a force pulling the rows of $Z_p$ towards these diverse prototypes. Assuming the input image contains diverse visual elements that activate a subset of these distinct prototypes, the feature matrix $Z_p$ is effectively regularized to span the same subspace as the active codebook vectors:

$$Z_p \xrightarrow{\mathcal{L}_{IVQ}} \text{span}(\{c_k\}_{active}) \subseteq \mathcal{S}_{code}. \tag{13}$$

Since the codebook maintains full rank ($\approx K$), it acts as a set of semantic anchors that prop open the feature space. This imposes an *implicit lower bound* on the feature rank, ensuring that representations maintain sufficient dimensionality to encode diverse semantic concepts rather than collapsing onto a single manifold.

**3. Structured vs. Indiscriminate Diversity.** This analysis also clarifies why IVQ outperforms general-purpose regularization methods like Barlow Twins or Spectral Regularization (as shown in Fig. 6).

- **General Regularization:** Methods that penalize cross-correlation or maximize spectral entropy encourage *indiscriminate diversity*. They force features to be orthogonal regardless of semantic content, which can lead to over-regularization where noise or irrelevant textures are amplified to satisfy the rank objective.

- **IVQ Regularization:** Our method fosters *structured diversity*. It preserves feature rank specifically along the semantic directions defined by the clinical concepts (the codebook). The feature space is allowed to be low-rank within a concept cluster (compressing intra-class variance) while maintaining high-rank separation between different concepts (preserving inter-class variance). This alignment between optimization dynamics and semantic structure is the key driver of IVQ-CBM's superior performance.

### C.4 THEORETICAL ANALYSIS FOR GRADIENT CONVERGENCE

The total loss function $\mathcal{L}_{\text{total}}$ is differentiable with respect to all trainable parameters $\theta$ of the proposed model. This ensures that gradients are well-defined, permitting stable model training via gradient-based optimizers.

*Proof.* The proof proceeds by analyzing the differentiability of each component of the total loss.

**Formal Definitions.** Let $\theta = \{\theta_E, \phi_A, \phi_f, \phi_Q\}$ denote the set of all trainable parameters, where $\theta_E$ are the parameters of the ViT-based visual encoder $\mathcal{E}$, $\phi_A$ are the learnable queries in the Magnet Attention module $\mathcal{A}$, $\phi_f$ are the parameters of the projection head $f$, and $\phi_Q$ is the learnable codebook of the Vector-Quantizer $\mathcal{Q}$.

The total loss is a weighted sum of its components:

$$\mathcal{L}_{\text{total}}(\theta) = \mathcal{L}_{\text{cls}} + \mathcal{L}_{\text{concept}} + \mathcal{L}_{\text{IVQ}}. \tag{14}$$

By the sum rule of differentiation, if each component loss is differentiable with respect to $\theta$, then $\mathcal{L}_{\text{total}}$ is also differentiable. We analyze each component in turn.

**Differentiability of the Main Prediction Path ($\mathcal{L}_{\text{cls}}$ and $\mathcal{L}_{\text{concept}}$).** The main prediction path computes visual concepts from patch embeddings $\mathbf{Z}_p = \mathcal{E}_{\theta_E}(\mathbf{x}) \in \mathbb{R}^{L \times D}$ and then calculates the classification and concept losses. The aggregated visual concepts, denoted $\mathbf{M} \in \mathbb{R}^{K \times D}$, are derived as:

$$\mathbf{M} = \mathcal{A}_{\phi_A}(\mathbf{Z}_p), \tag{15}$$

and the final logits are obtained after a projection head $f_{\phi_f}$. The Magnet Attention operator $\mathcal{A}_{\phi_A}$ involves computing squared Euclidean distances, applying a 'softmax' function, and performing a weighted sum via matrix multiplication. The projection head $f_{\phi_f}$ consists of standard neural network layers (e.g., Linear, LayerNorm). All these operations are continuously differentiable.

The losses $\mathcal{L}_{\text{cls}}$ and $\mathcal{L}_{\text{concept}}$ are computed using the cross-entropy function, which is smooth and differentiable with respect to its inputs. Therefore, by the chain rule, both losses are differentiable with respect to the parameters in their computational graphs, namely $\{\theta_E, \phi_A, \phi_f\}$.

**Differentiability of the VQ Regularization Path ($\mathcal{L}_{\text{IVQ}}$).** This part is critical as the Vector-Quantizer $\mathcal{Q}_{\phi_Q}$ contains a non-differentiable $\operatorname{argmin}$ operation for selecting the nearest codebook vector $\mathbf{e}_k \in \phi_Q$ for a given input patch feature $\mathbf{z}_j$:

$$k^* = \underset{k}{\operatorname{argmin}} \left\| \mathbf{z}_j - \mathbf{e}_k \right\|_2^2. \tag{16}$$

The derivative of this discrete selection is zero almost everywhere, which blocks gradient flow. The IVQ loss, however, is formulated to circumvent this issue:

$$\mathcal{L}_{\text{IVQ}} = \underbrace{\left\| \operatorname{sg}[\mathbf{Z}_p] - \mathcal{Q}_{\phi_Q}(\mathbf{Z}_p) \right\|_2^2}_{\text{Codebook Loss}} + \beta \cdot \underbrace{\left\| \mathbf{Z}_p - \operatorname{sg}[\mathcal{Q}_{\phi_Q}(\mathbf{Z}_p)] \right\|_2^2}_{\text{Commitment Loss}}, \tag{17}$$

where $\operatorname{sg}[\cdot]$ denotes the stop-gradient operator (equivalent to '.detach()').

We analyze the gradient of each component:

- **Gradient w.r.t. codebook $\phi_Q$:** The gradient for $\phi_Q$ flows only through the Codebook Loss term, as the Commitment Loss detaches the quantizer's output. The gradient is:

$$\nabla_{\phi_Q} \mathcal{L}_{\text{IVQ}} = \nabla_{\phi_Q} \left\| \operatorname{sg}[\mathbf{Z}_p] - \mathcal{Q}_{\phi_Q}(\mathbf{Z}_p) \right\|_2^2. \tag{18}$$

  This gradient is well-defined and updates the codebook vectors to move closer to the encoder's features.

- **Gradient w.r.t. encoder $\theta_E$:** The gradient for the encoder's parameters $\theta_E$ (which produce $\mathbf{Z}_p$) flows only through the Commitment Loss term. The gradient with respect to the encoder's output is:

$$
\begin{aligned}
\nabla_{\mathbf{Z}_p} \mathcal{L}_{\text{IVQ}} &= \beta \cdot \nabla_{\mathbf{Z}_p} \left\| \mathbf{Z}_p - \operatorname{sg}[\mathcal{Q}_{\phi_Q}(\mathbf{Z}_p)] \right\|_2^2 \\
&= 2\beta \left( \mathbf{Z}_p - \operatorname{sg}[\mathcal{Q}_{\phi_Q}(\mathbf{Z}_p)] \right).
\end{aligned} \tag{19}
$$

  This gradient is well-defined and is equivalent to that of a Mean Squared Error loss. It effectively pulls the encoder's output $\mathbf{Z}_p$ towards the selected (but detached) codebook vectors without passing through the non-differentiable $\operatorname{argmin}$ operation.

The use of the stop-gradient operator correctly decouples the updates, ensuring that computable gradients are available for both the encoder and the codebook.

**Conclusion.** We have established the differentiability of all components of the total loss function.

- The prediction losses, $\mathcal{L}_{\text{cls}}$ and $\mathcal{L}_{\text{concept}}$, are differentiable with respect to $\{\theta_E, \phi_A, \phi_f\}$.

- The regularization loss, $\mathcal{L}_{\text{IVQ}}$, provides well-defined gradients for both the codebook $\phi_Q$ and the encoder $\theta_E$.

Since $\mathcal{L}_{\text{total}}$ is a linear combination of these differentiable components, its gradient $\nabla_\theta \mathcal{L}_{\text{total}}$ is well-defined and can be computed via standard backpropagation. The existence of a valid gradient is a necessary condition for the convergence of gradient-based optimization algorithms. Therefore, the model architecture is theoretically sound for training. □

## D  IMPLEMENTATION DETAILS

**Experimental Setup.** To ensure a fair comparison, we reproduce all baselines within our experimental framework and adopt Biomedical-CLIP (Zhang et al., 2025) for our method and baselines. Based on this architecture, we insert a projection layer to map visual features to the textual feature space, thereby aligning their dimensions. We then apply `L2` normalization to regularize the aligned feature representations. We employ an exponential learning rate scheduler with a warm-up period and utilize the AdamW (Loshchilov & Hutter, 2017) optimizer for training. The initial learning rate and batch size are set to `1e-4` and `32`, respectively. All experiments are conducted using Python `3.9`, PyTorch `2.5.1`, and a single NVIDIA RTX `4090` GPU. We also conduct experiments within CLIP to validate the universality of representation collapse (Cherti et al., 2023).

**Concept Generation.** For the concepts associated with each dataset, we prompt Gemini 2.5 Pro to generate textual descriptions for each class. The generated concepts are then cross-validated using GPT-4o to ensure their quality and relevance. Drawing upon the work of (Panousis et al., 2024), we build a hierarchical framework for concepts based on a coarse-to-fine principle.

## E  ANALYSIS OF FEATURE RANK DYNAMICS

To empirically validate our method's ability to mitigate **representational collapse**, we analyze the dynamics of the feature rank during training. The rank of the feature matrix serves as a proxy for the diversity and richness of the learned representations. A sustained high rank indicates that the features are diverse and non-redundant, while a decline in rank—known as rank collapse—suggests that the feature space has become degenerate, hindering the model's ability to learn distinct concepts.

In this section, we present a series of experiments comparing our method against various baselines. The results consistently demonstrate the effectiveness of our approach in maintaining high-rank feature representations.

**Comparison with State-of-the-Art Baselines.** As illustrated in Figure 4, we first compare our model with several leading concept-based methods. Our approach (**Ours**) consistently maintains a high and stable feature rank across all eight datasets throughout the training process. In stark contrast, baseline methods such as Explicd, MVP-CBM, and others exhibit a noticeable decline in feature rank, succumbing to varying degrees of representational collapse. This result highlights our method's superior ability to preserve the expressive power of the feature space compared to existing approaches.

**Comparison on Standard Benchmarks.** As illustrated in Figure 5, we extend our analysis to standard vision benchmarks, ranging from CIFAR-10 to large-scale datasets like ImageNet. Consistent with our observations in the medical domain, our approach (**Ours**) maintains a robust and stable feature rank throughout training across all five datasets. In stark contrast, baseline methods succumb to varying degrees of representational collapse, a phenomenon that becomes notably more severe on complex datasets. Specifically, MVP-CBM and DOT-CBM exhibit a drastic decline in feature rank on ImageNet and Places365, indicating a failure to preserve feature diversity at scale. This empirical evidence confirms that representational collapse is a fundamental bottleneck in CBMs,

and highlights our method's superior ability to generalize and preserve expressive power even in large-scale classification scenarios.

**Ablation Study on the IVQ Module.** To isolate the contribution of our proposed IVQ module, we conduct a crucial ablation study, with the results presented in Figure 6. The comparison is stark: the model equipped with our IVQ module (`w/ IVQ`, solid lines) successfully sustains a high feature rank on both training and validation sets. Conversely, the model without it (`w/o IVQ`, dashed lines) experiences a sharp drop in rank, mirroring the behavior of the baseline models. This provides compelling evidence that the IVQ module is the key component responsible for preventing feature space degeneracy and maintaining representational diversity.

**Comparison with Representation Regularization Techniques.** Furthermore, we extend our analysis to include other common representation regularization techniques in Figure 7. While some regularization methods may also help in maintaining a higher feature rank, it is crucial to note a key insight: **a high feature rank is a necessary, but not sufficient, condition for superior performance.** Simply forcing features to be diverse (e.g., via decorrelation penalties) does not guarantee that they are meaningful or well-aligned with the downstream task. As shown in the figure, our method not only preserves rank effectively but also achieves this in a way that structures the feature space for better concept learning, ultimately leading to improved overall performance (as shown in section 4.2). This distinguishes our approach from methods that might artificially inflate rank without enhancing semantic representation.

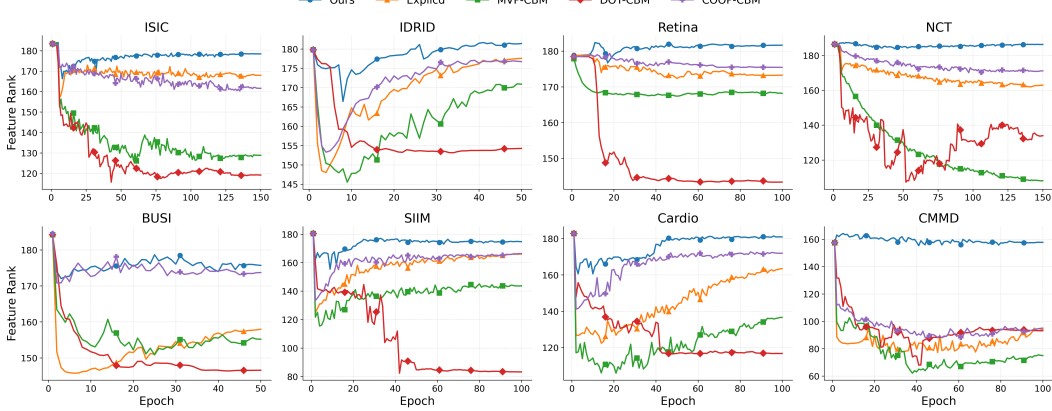

Figure 4: Comparative analysis of feature rank dynamics across eight datasets. Our proposed method (**Ours**) successfully maintains a high and stable feature rank, while baseline methods, including Explicd, MVP-CBM, DOT-CBM, and COOP-CBM, exhibit varying degrees of rank collapse during training.

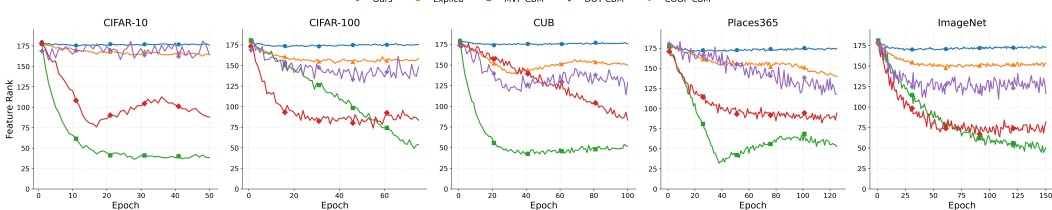

Figure 5: Comparative analysis of feature rank dynamics across five general domain datasets. Our proposed method (**Ours**) successfully maintains a high and stable feature rank, while baseline methods, including Explicd, MVP-CBM, DOT-CBM, and COOP-CBM, exhibit varying degrees of rank collapse during training.

# F VISUALIZATION OF CODEBOOK IN IVQ

To qualitatively evaluate the semantic knowledge captured by the IVQ codebook, we provide a comprehensive visual analysis across all six medical imaging datasets. In the following figures, we

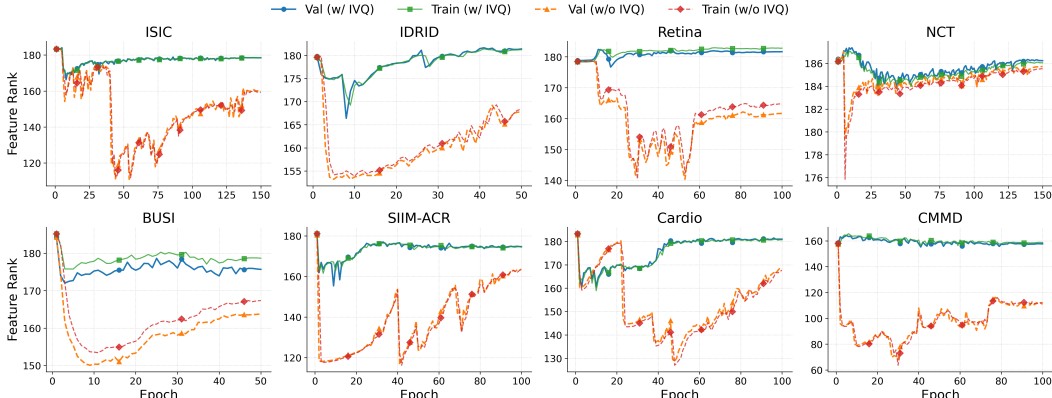

Figure 6: Feature rank dynamics for models with (`w/ IVQ`, solid lines) and without (`w/o IVQ`, dashed lines) our IVQ module, shown on both training and validation sets. The clear gap demonstrates the effectiveness of IVQ in preventing rank collapse.

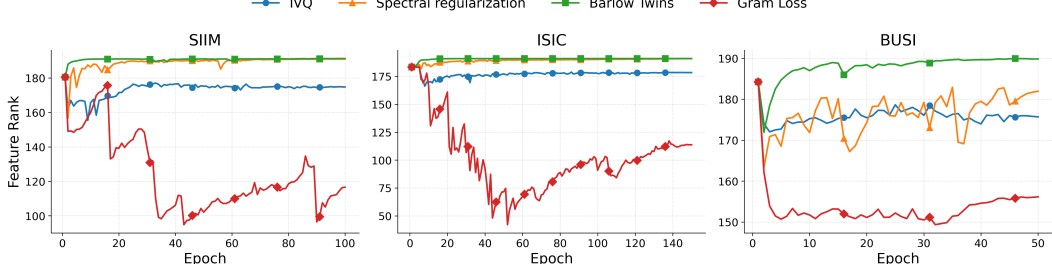

Figure 7: Feature rank dynamics of IVQ against regularization baselines on the SIIM, ISIC, and BUSI datasets. IVQ (blue) and Barlow Twins (green) successfully maintain a high rank, while Spectral regularization (orange) is unstable and Gram Loss (red) suffers a severe collapse.

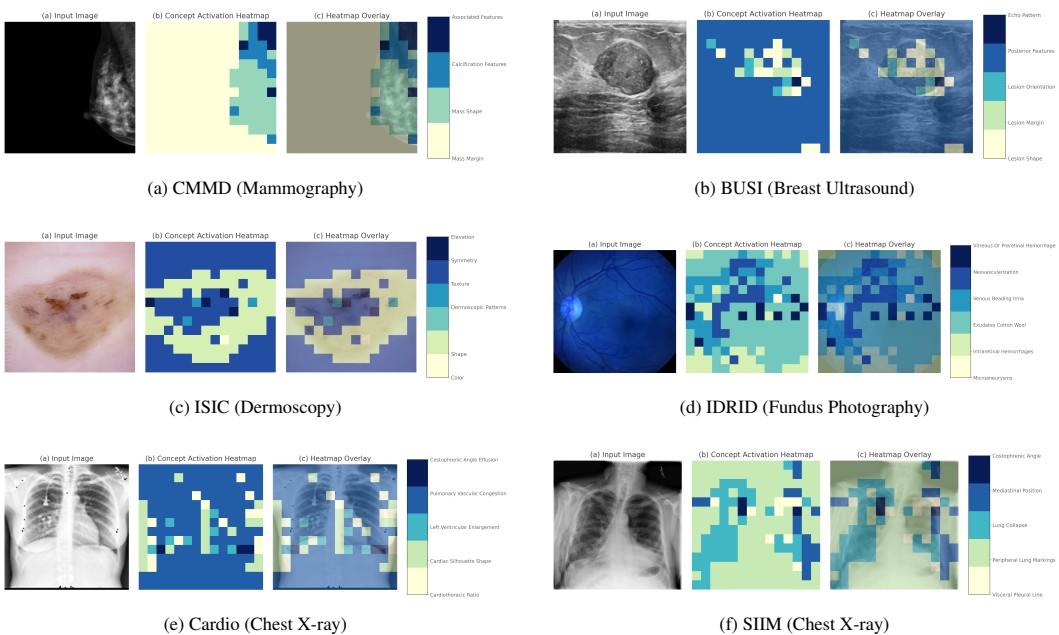

Figure 8: Visualization of the learned codebook prototype mappings across six medical imaging datasets. For each dataset, we show (a) the input image, (b) the concept activation heatmap where each color corresponds to the most active prototype for that patch, and (c) the overlay. The color legend maps each prototype to a human-defined clinical concept.

visualize the prototype index maps for representative images. For each image patch, we identify the nearest codebook prototype via an argmin operation and color-code the patch according to that prototype's index. This process generates a concept activation heatmap, revealing which learned visual prototype is most dominant in each region of the image. The results consistently demonstrate that our model learns semantically meaningful and clinically relevant concept representations that align with the diagnostic reasoning of medical experts.

**Analysis on CMMD (Mammography).** As shown in Figure 8a, the resulting mapping demonstrates a highly logical process that mirrors clinical reasoning. The prototype for `Mass Margin` precisely delineates the lesion's contour, while those for `Mass Shape` and `Calcification Features` correspond to its internal characteristics. Crucially, the prototype for `Associated Features` extends its focus beyond the lesion's border, probing the surrounding parenchyma for signs of structural distortion—a key indicator of malignancy.

**Analysis on BUSI (Breast Ultrasound).** The visualization for BUSI in Figure 8b reveals a structured analysis of the breast lesion. The prototypes for `Lesion Shape` and `Lesion Margin` accurately capture the overall form and boundary of the hypoechoic mass. Progressing to finer details, the prototypes for `Posterior Features` and `Echo Pattern` are activated on the internal and posterior regions of the lesion, which are critical assessments in the BI-RADS classification system.

**Analysis on ISIC (Dermoscopy).** In the dermoscopy example Figure 8c, the prototype mappings align well with clinical assessment criteria like the "ABCDE" rule. Prototypes for general attributes such as `Shape` and `Color` are active across the broader lesion area. Meanwhile, more specific and clinically crucial prototypes, like `Dermoscopic Patterns` and `Texture`, are correctly localized to the darker, diagnostically significant interior regions.

Most importantly, this visualization addresses the critical issue of concept ambiguity (Kim et al., 2023), where a given image may lack a specific concept entirely—a key challenge for concept-based models. For instance, while this lesion has a distinct texture, it may not exhibit `Streaks`. Our model correctly reflects this by showing negligible activation for the corresponding prototype.

This demonstrates a faithful alignment, as the model does not erroneously force a prediction for a feature that is not visually present, confirming that our learned codebook achieves a truly accurate and discerning concept assignment.

**Analysis on IDRID (Fundus Photography).** The analysis of the diabetic retinopathy case in Figure 8d shows that the model differentiates between various pathologies. While general findings like `Intraretinal Hemorrhages` are mapped to wider areas, the prototypes for severe, high-risk pathologies such as `Neovascularization` and `Vitreous Hemorrhage` are correctly concentrated near critical anatomical structures like the optic disc and major vascular arcades.

**Analysis on Cardio & SIIM (Chest X-ray).** The chest X-ray visualizations Figure 8e and Figure 8f) demonstrate a strong anatomical grounding. Prototypes for global assessments like `Cardiothoracic Ratio` and `Cardiac Silhouette Shape` are broadly active over the heart. In contrast, prototypes for specific pathologies are precisely localized. For instance, `Pulmonary Vascular Congestion` is mapped to the lung fields, while `Costophrenic Angle Effusion` is correctly activated in the lower lobes of the lungs where pleural fluid accumulates. This showcases a spatially aware reasoning process that distinguishes between global shape and localized pathological signs.

**Codebook Visualization Reveals Structured and Disentangled Concepts.** To qualitatively assess the structure of the learned codebook, we visualize the code vectors using UMAP by projecting them into a 3D space. As illustrated in Figure 9, a clear and consistent pattern emerges across all six distinct datasets. The learned codes for each dataset are well-separated, forming discrete and compact clusters with significant distance between them. This spatial separation is highly desirable, as it indicates that the learned codes are disentangled and non-redundant. Each code has successfully converged to represent a unique, semantically distinct concept, avoiding representational collapse where multiple codes might capture similar features. The consistency of this structured outcome across diverse medical imaging modalities—from ultrasound (BUSI) and mammography (CMMD) to dermoscopy (ISIC) and beyond—demonstrates the robustness and generalizability of our method in discovering a meaningful basis of concepts. The formation of such a clean and well-structured codebook is foundational to the model's ability to make interpretable and reliable predictions.

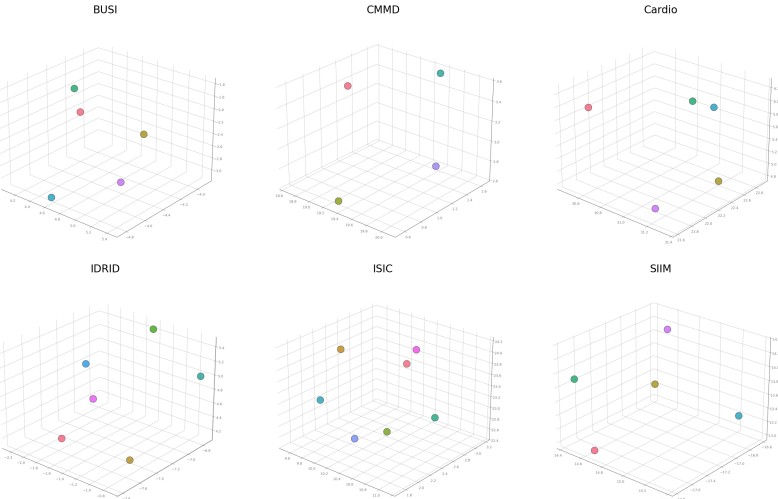

Figure 9: UMAP visualization for the codebook in various datasets. The clear spatial separation between codes in each plot indicates a highly disentangled and non-redundant set of learned concepts. This consistent structure across all domains demonstrates the robustness of our method in discovering a meaningful conceptual basis.

## G  VISUALIZATION OF CODEBOOK IN VQ

For comparison, we visualize the prototype mappings learned by a standard VQ baseline in Figure 10. This ablation study qualitatively demonstrates the limitations of standard VQ, which

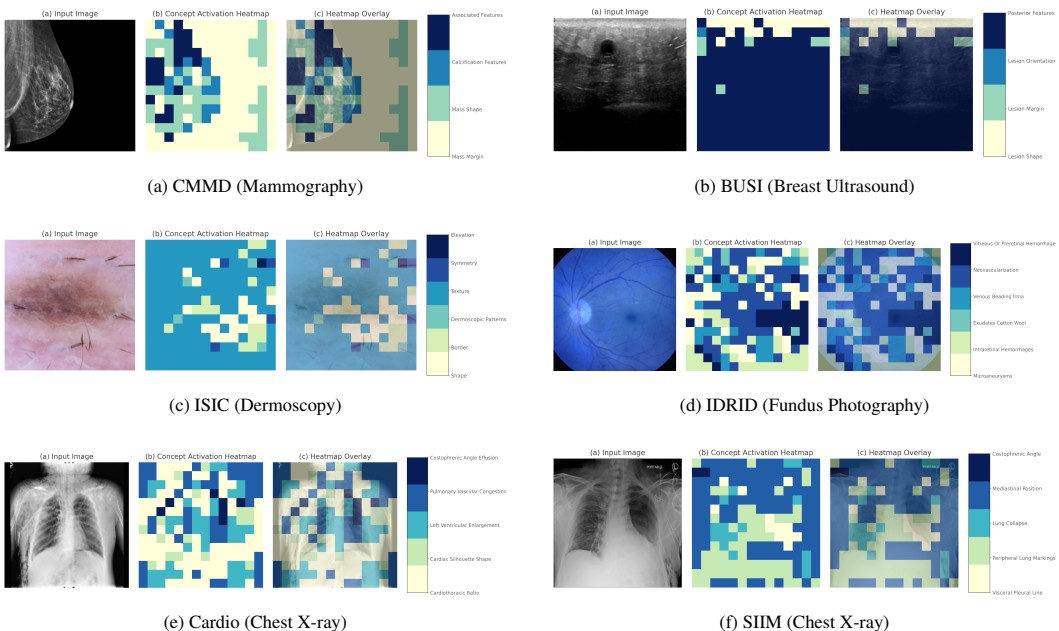

Figure 10: Visualization of the learned codebook prototype mappings from the standard VQ baseline. These mappings highlight a significant failure in capturing clinically-relevant concepts. Prototypes are often scattered, misaligned with anatomical structures, and fail to differentiate between pathology and background, starkly contrasting with the precise mappings of our IVQ model (shown in Figure 8).

struggles to learn semantically meaningful and spatially-precise concepts. In stark contrast to the clinically-aligned prototypes from our proposed IVQ method, the VQ baseline's mappings often reveal a significant lack of anatomical grounding. The resulting heatmaps show concepts activated in nonsensical locations, failing to distinguish between critical pathologies and irrelevant background, thus highlighting the necessity of our IVQ approach.

**Analysis on CMMD (Mammography).** The VQ mapping in Figure 10a fails to capture any meaningful anatomical structure. The prototype for `Mass Margin` (yellow) is nonsensically activated in the background adipose tissue rather than on the lesion's contour. Similarly, `Mass Shape` (light green) and `Calcification Features` (blue) are scattered randomly, failing to delineate the actual lesion or its internal characteristics. This mapping lacks the logical, clinically-relevant precision required for diagnosis.

**Analysis on BUSI (Breast Ultrasound).** The BUSI visualization (Figure 10b) demonstrates a near-total failure of the VQ model. The actual hypoechoic lesion is almost entirely ignored or misclassified as the `Posterior Features` (dark blue) prototype, which dominates the entire image background. Critical concepts like `Lesion Shape` and `Lesion Margin` are only sparsely and incorrectly activated in the surrounding tissue, showing no understanding of the target pathology.

**Analysis on ISIC (Dermoscopy).** This mapping (Figure 10c) shows poor conceptual differentiation. The prototype for `Dermoscopic Patterns` (light green) incorrectly dominates almost the entire image, including the clear background skin, indicating it has not learned a specific feature. Furthermore, `Border` and `Shape` prototypes are activated illogically inside the lesion rather than at its periphery, reversing the correct diagnostic process.

**Analysis on IDRID (Fundus Photography).** The VQ mapping for IDRID (Figure 10d) is chaotic and lacks anatomical precision. High-risk pathologies like `Neovascularization` and `Venous Beading` are scattered randomly across the retina, failing to co-localize with critical structures like the optic disc or major vascular arcades. This all-over-the-place activation suggests the model has only learned coarse pixel statistics rather than a true, spatially-aware understanding of the pathology.

**Analysis on Cardio & SIIM (Chest X-ray).** The chest X-ray visualizations demonstrate severe anatomical flaws. In the Cardio example (Figure 10e), the prototype for `Cardiothoracic Ratio` (yellow) bleeds nonsensically into the lung fields and abdomen. Critically, `Costophrenic Angle Effusion` (dark blue) is activated in the *upper* lung zones, which is clinically impossible as fluid accumulates at the lung bases. Similarly, in the SIIM image (Figure 10f), the `Costophrenic Angle` (dark blue) is again misplaced in the upper chest, and the `Visceral Pleural Line` prototype fails to trace the actual line of the collapsed lung.

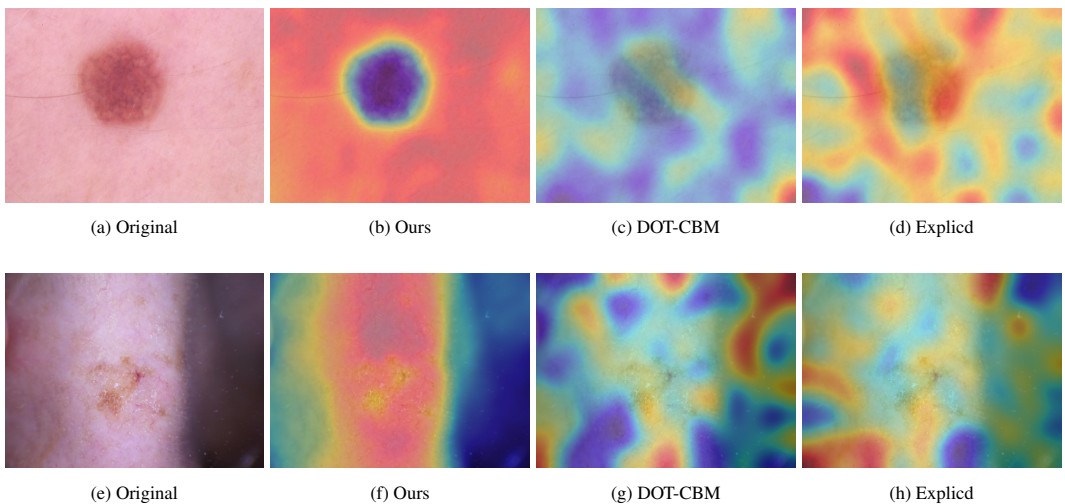

(a) Original     (b) Ours     (c) DOT-CBM     (d) Explicd

(e) Original     (f) Ours     (g) DOT-CBM     (h) Explicd

Figure 11: **Qualitative comparison of explanation heatmaps on ISIC 2018 dermoscopy images (skin lesion classification).** We visualize concept attributions using Grad-CAM-style activation maps (Selvaraju et al., 2019) applied to the concept bottleneck layer (or post-hoc concept predictor) of each method, overlaid on the original images with jet colormap and 40% opacity. Our method consistently produces sharp, highly localized activations that precisely delineate clinically relevant lesion regions while effectively suppressing background skin and artifacts. In contrast, DOT-CBM and Explicd generate diffuse, noisy, or poorly localized patterns that frequently spill into surrounding healthy skin or fail to adequately highlight diagnostic features of the lesion.

## H VISUALIZATION ANALYSIS WITH BASELINES

To empirically evaluate the faithfulness and localization quality of the learned concepts, we conduct a qualitative comparison between our proposed IVQ-CBM and two recent patch-based baselines: DOT-CBM and Explicd. We employ Grad-CAM (Selvaraju et al., 2019) to generate concept activation maps, visualizing the spatial regions that contribute most significantly to the model's concept predictions. This analysis is performed across two distinct domains: medical imaging (ISIC 2018) and fine-grained object classification (CUB-200-2011).

**Medical Imaging (ISIC 2018).** Figure 11 presents the visualization results on dermoscopy images. The comparison reveals a stark contrast in the semantic coherence of the learned features. Our method (Columns (b) and (f)) produces sharp, object-centric activations that precisely delineate the lesion boundaries, effectively separating the pathological tissue from healthy skin. In contrast, the baselines exhibit severe visual degradation indicative of representational collapse. DOT-CBM (Columns (c) and (g)) displays diffuse, "cloudy" activation patterns that spill significantly into the background, suggesting a failure to disentangle foreground concepts from noise. Similarly, Explicd (Columns (d) and (h)) suffers from scattered and disjointed activations that often miss the lesion center entirely. These visual artifacts confirm that without explicit regularization to maintain feature rank, the resulting concept representations become informationally redundant and spatially ambiguous.

**General Object Classification (CUB-200-2011).** To demonstrate generalizability, we extend this analysis to the CUB-200-2011 dataset in Figure 12. Consistent with the medical domain, our approach generates well-localized heatmaps that cover discriminative avian parts (e.g., head, wings, torso) while suppressing complex background clutter. Conversely, the baselines struggle with lo-

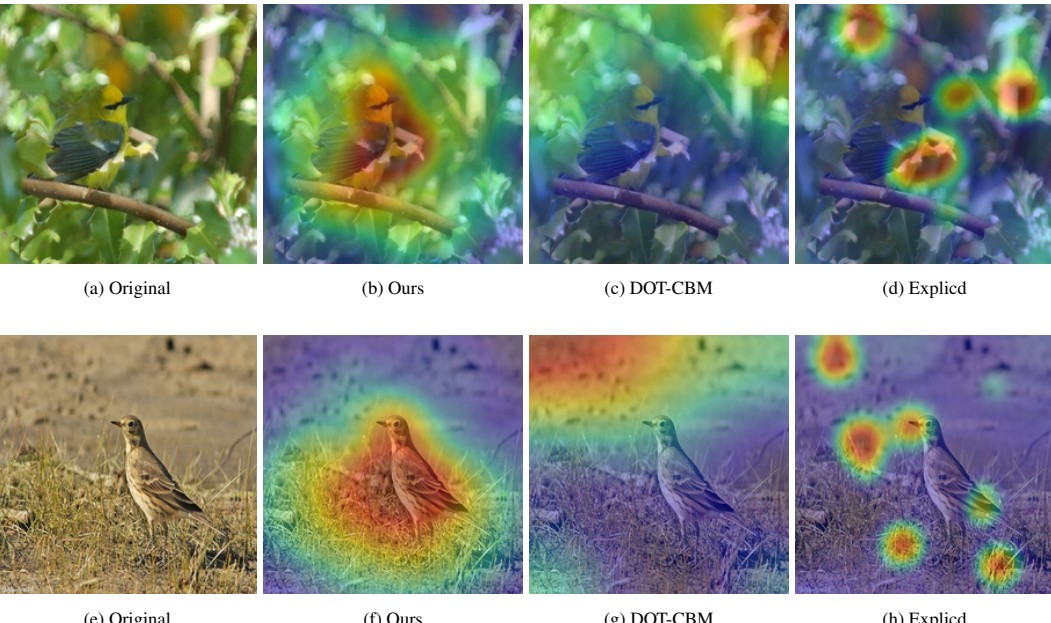

Figure 12: **Qualitative comparison of explanation heatmaps on the CUB-200-2011 dataset (fine-grained bird classification).** We visualize concept attributions using Grad-CAM-style activation maps (Selvaraju et al., 2019) applied to the concept bottleneck layer (or post-hoc concept predictor) of each method, overlaid on the original images with jet colormap and 40% opacity. Our approach consistently generates sharp, well-localized activations that faithfully highlight the entire discriminative bird regions (head, breast, wings, and tail) while effectively suppressing irrelevant background. By contrast, DOT-CBM and Explicd produce diffuse, over-smoothed, or noisy activation patterns that either leak into background areas or fail to adequately cover diagnostically relevant bird parts.

calization; their activations are either over-smoothed or erroneously highlight background elements (e.g., branches or ground).

**Conclusion.** The superior visualization quality of IVQ-CBM—characterized by precise localization and boundary adherence—is not merely a result of better training dynamics but a direct phenomenological manifestation of our high-rank feature space. By successfully escaping the low-rank trap via Implicit Vector Quantization, our model preserves the semantic diversity required for faithful and interpretable visual concept learning.

# I  RESULTS WITH OTHER MULTI-MODAL BACKBONE

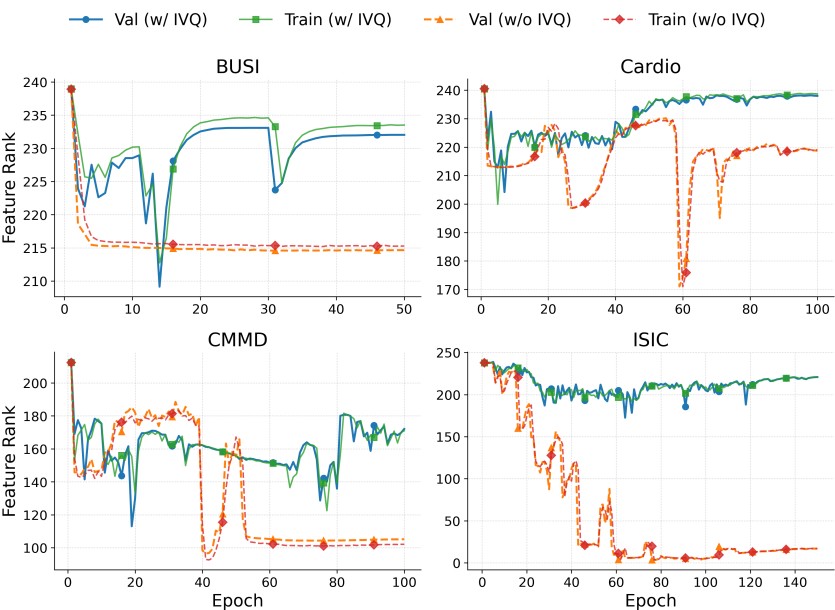

Figure 13: Ablation study on the impact of the IVQ component. The figure compares the feature rank dynamics of the model trained with and without the IVQ module with CLIP backbone, demonstrating its effect on representation diversity.

Our primary experiments are conducted on a CLIP model pre-trained with a biomedical corpus (Zhang et al., 2025) to build the proposed IVQ-CBM. This domain-specific backbone ensures feature relevance for medical tasks. Within this main setup, we perform comprehensive ablation studies to validate the contribution of each component. Figure 6 provides a compelling visualization of a crucial ablation study on the IVQ module. The results are unequivocal: the model variant without IVQ (dashed lines) suffers a significant drop in feature rank during training, a clear indication of representational collapse. In stark contrast, the full model equipped with IVQ (solid lines) robustly maintains a high feature rank throughout the training process. Notably, on datasets such as BUSI and CMMD, it even exhibits rank recovery after initial dips. This direct comparison underscores the indispensable role of IVQ in preserving feature space dimensionality and preventing catastrophic rank collapse.

A key question is whether the observed rank collapse is specific to the domain-adapted biomedical encoder or if it represents a more general challenge in representation learning. To investigate this, we replicated the experiment using a general-purpose feature extractor. Specifically, we replaced the specialized backbone with a standard pre-trained Vision Transformer (ViT-L/14) (Radford et al., 2021), with weights from the OpenCLIP project (Cherti et al., 2023) trained on the LAION-2B dataset (Schuhmann et al., 2022). As demonstrated in Figure 13, the results on four distinct datasets are unequivocal. The models trained without our IVQ module (dashed lines) consistently suffer from a severe drop in feature rank, confirming that representational collapse is not tied to a specific encoder. In stark contrast, the models equipped with IVQ (solid lines) robustly maintain a high and stable feature rank throughout training. This confirms that rank collapse is a fundamental problem and establishes IVQ as a robust, model-agnostic solution for preserving representation diversity.

## J    STATEMENT ON THE USE OF LARGE LANGUAGE MODELS

Throughout the preparation of this manuscript, we utilized large language models to enhance the quality of the text. Specifically, we employed Google's Gemini Pro for tasks related to language refinement, including correcting grammar and spelling, improving sentence clarity, and ensuring a consistent academic tone.

The core scientific contributions, including the formulation of the problem, the proposed methodology, the design and execution of experiments, and the interpretation of results, are entirely the work of the authors. All text generated or modified by the LLM was meticulously reviewed, edited, and revised by the authors to ensure it accurately reflects our original ideas and findings. The authors bear full and final responsibility for all content presented in this paper.

## K    PSEUDOCODE OF IVQ-CBM

---

**Algorithm 1** Training Procedure of Our Proposed Method

---

**Require:** Training dataset $\mathcal{D} = \{(x, \mathbf{c}, y)\}$.
**Require:** Visual encoder $E_I$, text encoder $E_t$, classifier head $h_{\text{cls}}$.
**Require:** Learnable concept queries $\mathbf{Q} \in \mathbb{R}^{K \times D}$.
**Require:** Learnable IVQ codebook $\mathcal{C}_{vq} \in \mathbb{R}^{K \times D}$.
**Require:** Commitment cost hyperparameter $\beta$.
**Require:** Set of textual concept descriptions $\{t_k\}_{k=1}^{K}$.
 1: Initialize parameters for $E_I$, $h_{\text{cls}}$, $\mathbf{Q}$, and $\mathcal{C}_{vq}$.
 2: Pre-compute text concept embeddings: $\boldsymbol{\tau}_k \leftarrow E_t(t_k)$ for $k = 1, \ldots, K$.
 3: **for** each training epoch **do**
 4:     **for** each batch $(x, \mathbf{c}, y)$ sampled from $\mathcal{D}$ **do**
 5:                                                               ▷ **1. Visual Feature Extraction**
 6:         $\mathbf{Z}_v = [\mathbf{z}_{\text{cls}}, \mathbf{z}_1, \ldots, \mathbf{z}_L] \leftarrow E_I(x)$               ▷ Encode image
 7:         $\mathbf{Z}_p \leftarrow [\mathbf{z}_1, \ldots, \mathbf{z}_L]$                      ▷ Extract patch-level features
 8:                                                               ▷ **2. Magnet Concept Aggregation**
 9:         Compute soft-assignment matrix $\mathbf{A} \in \mathbb{R}^{L \times K}$ between patches and queries:
10:         $A_{jk} \leftarrow \frac{\exp(-\|\mathbf{z}_j - \mathbf{q}_k\|_2^2)}{\sum_{k'=1}^{K} \exp(-\|\mathbf{z}_j - \mathbf{q}_{k'}\|_2^2)}$                    ▷ For each patch $j$ and query $k$
11:         $\mathcal{M} \leftarrow \mathbf{A}^\top \mathbf{Z}_p$                        ▷ Aggregate features into $K$ Visual Concepts
12:                                                               ▷ **3. Concept-Text Alignment**
13:         $\mathbf{v} \leftarrow \text{diag}(\mathcal{M}\boldsymbol{\tau}^\top)$           ▷ Generate Concept Activation Vector (CAV)
14:                                                               ▷ **4. Final Classification**
15:         $\mathbf{p} \leftarrow h_{\text{cls}}(\mathbf{v})$                       ▷ Predict class logits from CAV
16:                                                               ▷ **5. Loss Computation**
17:         $\mathcal{L}_{\text{cls}} \leftarrow \mathcal{L}_{\text{CE}}(\mathbf{p}, y)$                ▷ Cross-entropy loss for task classification
18:         $\mathcal{L}_{\text{concept}} \leftarrow \mathcal{L}_{\text{BCE}}(\mathbf{v}, \mathbf{c})$               ▷ BCE loss for concept supervision
19:                                   ▷ – Implicit Vector Quantization (IVQ) Regularization –
20:         For each patch feature $\mathbf{z}_j \in \mathbf{Z}_p$, find nearest codebook vector $\mathbf{c}_{k_j} \in \mathcal{C}_{vq}$.
21:         Let $\mathbf{Z}_q = [\mathbf{c}_{k_1}, \ldots, \mathbf{c}_{k_L}]$ be the matrix of quantized features.
22:         $\mathcal{L}_{\text{IVQ}} \leftarrow \|\text{sg}(\mathbf{Z}_p) - \mathbf{Z}_q\|_2^2 + \beta\|\mathbf{Z}_p - \text{sg}(\mathbf{Z}_q)\|_2^2$              ▷ IVQ loss
23:                                                     ▷ – Overall Objective and Optimization –
24:         $\mathcal{L} \leftarrow \mathcal{L}_{\text{cls}} + \mathcal{L}_{\text{concept}} + \mathcal{L}_{\text{IVQ}}$                     ▷ Total loss
25:         Update parameters of $E_I, h_{\text{cls}}, \mathbf{Q}, \mathcal{C}_{vq}$ using gradients from $\mathcal{L}$.
26:     **end for**
27: **end for**

---

## L    QUANTITATIVE INTERPRETABILITY AND CAUSAL FAITHFULNESS

While qualitative visualizations offer intuitive insights, rigorous validation requires quantifying how well the learned concepts align with human expertise and whether they causally drive the model's

Table 6: **Quantitative Human Evaluation Results.** Two radiologists evaluated the interpretability of IVQ-CBM on localization quality (Likert Scale 1–5) and semantic consistency (Choice Accuracy).

| Dataset | Task 1: Localization Accuracy (Avg. Likert Score, 1–5) | Task 2: Semantic Alignment (Avg. Choice Accuracy) |
|---|---|---|
| CMMD (Mammography) | 4.32 | 90% (45/50) |
| BUSI (Ultrasound) | 4.51 | 92% (46/50) |
| ISIC (Dermoscopy) | 4.15 | 88% (44/50) |
| IDRID (Fundus Photography) | 4.20 | 86% (43/50) |
| **Average** | **4.30** | **89%** |

decisions. To this end, we conducted two supplementary studies: a human evaluation with radiologists and a concept intervention test.

### L.1 HUMAN EVALUATION WITH EXPERT RADIOLOGISTS

To assess the spatial and semantic alignment of our learned codebook prototypes with clinical knowledge, we invited two radiologists to participate in a blinded evaluation study.

**Experimental Protocol.** We randomly sampled 50 images across four diverse datasets: CMMD (mammography), BUSI (ultrasound), ISIC (dermoscopy), and IDRID (fundus photography). The evaluation comprised two distinct tasks:

1. **Concept Localization (Spatial):** Experts were presented with an image and a target clinical concept (e.g., *Mass Margin*). They rated the model-generated activation heatmap on a 5-point Likert scale ($1$ = Completely Misaligned, $5$ = Perfectly Aligned) based on whether the highlighted region corresponded to the clinical pathology.

2. **Semantic Alignment (Conceptual):** Experts viewed the image and the heatmap *without* the concept label. They were asked to identify the represented concept from a multiple-choice list containing the ground truth and 3–4 plausible semantic distractors (e.g., distinguishing *Calcification* from *Mass Margin*).

**Results.** As detailed in Table 6, IVQ-CBM achieved a high average localization score of **4.30/5.0** and a semantic alignment accuracy of **89%**. These results quantitatively corroborate that our IVQ mechanism successfully anchors visual features to semantically meaningful and spatially accurate clinical concepts.

Table 7: **Concept Intervention Test on CMMD.** We measure the drop in the model's predicted probability for the "Malignant" class after manually intervening on specific concept activations. The substantial drop confirms the causal role of these concepts in the decision process.

| Intervention Type | Avg. $P(\text{Malignant})$ | Relative Change |
|---|---|---|
| None (Baseline) | 0.96 | – |
| Intervene on "Mass Margin" | 0.80 | -16.7% |
| Intervene on "Mass Shape" | 0.85 | -11.5% |
| **Intervene on Both** | **0.72** | **-25.0%** |

### L.2 CAUSAL FAITHFULNESS VIA CONCEPT INTERVENTION

A critical requirement for CBMs is *causal faithfulness*—meaning the concepts should not merely correlate with the prediction but actively drive it. We validated this via a counterfactual intervention test on the CMMD dataset.

**Methodology.** We selected a test subset of 50 images where the model confidently predicted malignancy ($P(\text{Malignant}) > 0.9$). We identified clinically causal concepts for this decision, specifically *Spiculated Margin* and *Irregular Shape*. We then performed an intervention on the Concept Activation Vector (CAV), denoted as $v$. Specifically, we "flipped" the activation scores of these malignant

Table 8: Evaluation of IVQ-CBM on the ANEC-5 Metric. The results demonstrate minimal performance degradation when restricting the model to the top-5 effective concepts, indicating robustness against information leakage.

| Dataset | Domain | Full Model Acc (%) | ANEC-5 Acc (%) | Performance Drop ($\Delta$) |
|---------|--------|--------------------|----------------|------------------------------|
| **CUB** | Natural | 75.93 | 75.21 | -0.72% |
| **CIFAR-10** | Natural | 87.92 | 86.85 | -1.07% |
| **ISIC** | Medical | 90.11 | 89.20 | -0.91% |
| **BUSI** | Medical | 93.59 | 92.80 | -0.79% |

concepts by replacing them with the average scores derived from benign samples, while keeping the classification head $h_{cls}$ frozen. We then measured the degradation in the model's predicted probability for the malignant class.

**Results.** Table 7 demonstrates the impact of these interventions. Modifying individual concepts resulted in a notable decrease in confidence. Crucially, intervening on both shape and margin concepts caused a substantial **25.0% drop** in the predicted probability (from 0.96 to 0.72). This significant sensitivity confirms that the decision-making logic of IVQ-CBM causally relies on these high-level clinical concepts, rather than spurious correlations or background artifacts.

## M ANALYSIS OF INFORMATION LEAKAGE AND EFFECTIVE CONCEPT LEARNING

Recent studies (Yan et al., 2023; Srivastava et al., 2025) have highlighted a critical challenge in CBM: the potential for information leakage in the bottleneck layer, where models may achieve high accuracy by encoding non-semantic noise rather than learning meaningful concepts. To rigorously validata the faithfulness of our approach, we evaluate IVQ-CBM using the A-NEC (Accuracy at Number of Effective Concepts) metric proposed by Srivastava et al. (2025) and provide a theoretical analysis of the mechanisms inherent to our architecture that mitigate such leakage.

### M.1 ROBUSTNESS EVALUATION ON THE A-NEC METRIC

The A-NEC metric assesses whether a model relies on a concise set of "effective" concepts or exploits a diffuse sum of features (leakage). We conducted an evaluation on both natural (CUB, CIFAR-10) and medical (ISIC, BUSI) datasets. Specifically, we performed post-hoc pruning on the final classification layer, retaining only the top-5 contributing concepts for each class prediction (denoted as ANEC-5), and compared this restricted performance against that of the full model.

As presented in Table 8, IVQ-CBM retains over 98% of its original performance across all datasets even when restricted to utilizing only the top-5 concepts. This minimal performance drop indicates that the model's decision-making is primarily driven by a few highly relevant semantic concepts, rather than relying on residual leakage distributed across irrelevant dimensions.

### M.2 MECHANISM ANALYSIS: MITIGATING LEAKAGE VIA STRUCTURAL REGULARIZATION

While prior approaches mitigate leakage by enforcing sparsity constraints on classifier weights, IVQ-CBM addresses the root cause—feature representation—through two complementary mechanisms:

**IVQ as a Semantic Filter.** Information leakage often exploits high-frequency noise or low-rank degenerate subspaces that carry discriminative but non-semantic information. The proposed IVQ mechanism acts as a structure-inducing regularizer. By imposing the Commitment Loss (Eq. 4), continuous patch features are compelled to align with learned codebook prototypes, which function as semantic anchors. This process effectively filters out unstructured noise, as leakage patterns typically lack the statistical consistency to form stable clusters within the codebook. By enforcing semantic consistency with discrete prototypes, IVQ prevents the bottleneck from serving as a generic conduit for pixel-level noise.

Table 9: Performance comparison on the ISIC dataset before and after concept refinement. Guided by the low assignment frequency in the IVQ codebook, replacing the dormant concept *Rough Texture* with *Irregular Streaks* leads to consistent gains.

| Model Setting | Concept Configuration | ACC (%) | BMAC (%) |
|---|---|---|---|
| Initial | Includes *Rough Texture* | 90.11 | 86.22 |
| Refined | Includes *Irregular Streaks* | **90.75** | **87.12** |
| *Performance Gain* | | +0.64 | +0.90 |

**Magnet Attention as Implicit Sparsity.** The Magnet Attention mechanism aggregates features via a competitive Softmax dynamic. This introduces an *implicit sparsity* effect: patches must "vote" strongly for specific concept queries to be aggregated. Diffuse background noise or leakage, which typically manifests as weak, uniform signals across patches, is naturally suppressed by the exponential nature of the Softmax function. Consequently, the resulting Visual Concepts $\mathcal{M}$ are composed of strong, concept-aligned signals, effectively achieving the goal of effective concept learning during the feature aggregation stage.

### M.3 IVQ Codebook as a Diagnostic Tool for Concept Refinement

Beyond its role as a regularizer, the IVQ codebook offers a unique capability: it functions as a transparent diagnostic tool to audit the quality of predefined textual concepts. Since the commitment loss (Eq. 4) compels codebook vectors to serve as semantic anchors for visual patches, we hypothesize that the frequency of patch assignments to each prototype serves as a quantitative proxy for that concept's visual validity.

To validate this, we conducted a post-hoc analysis of prototype assignment statistics on the ISIC dataset. The analysis revealed a stark dichotomy: while robust concepts such as *Blue-whitish veil* attracted tens of thousands of patch assignments, the concept *Rough Texture* received negligible attention ($< 50$ assignments across the entire validation set). This quantitative signal suggests that *Rough Texture* is a tactile property ill-suited for 2D dermoscopy classification, effectively rendering it a null concept in the visual domain.

Leveraging this insight, we implemented a closed-loop refinement process summarized in Table 9. We replaced the ineffective concept with a more visually distinct alternative, *Irregular Streaks*, and retrained the model. As shown in the table, this targeted adjustment yielded a tangible performance improvement, boosting Accuracy to **90.75%** (+0.64%) and BMAC to **87.12%** (+0.90%). This experiment demonstrates that IVQ-CBM provides a mechanism not only for learning concepts but for validating and refining the human knowledge base itself.

### N Sensitivity Analysis of Commitment Cost Parameter

The commitment cost hyperparameter $\beta$ in the IVQ loss (Eq. 4) serves a critical role in balancing the strength of the regularization, specifically controlling how tightly the visual encoder's output is constrained to the learnable codebook prototypes. To empirically validate the model's sensitivity to this design choice, we conducted ablation experiments on three diverse datasets: ISIC (Dermoscopy), SIIM (Chest X-ray), and BUSI (Ultrasound), evaluating a standard range of values $\beta \in \{0.1, 0.25, 0.5, 1.0\}$.

As summarized in Table 10, the model demonstrates remarkable robustness to variations in $\beta$. The performance fluctuations in both Accuracy and BMAC are negligible (typically $< 0.5\%$) across all tested values. This stability suggests that the primary function of the IVQ loss is to provide a structural constraint that prevents feature collapse. Our empirical results indicate that within the examined range, the model is not sensitive to the precise magnitude of $\beta$, suggesting that the regularization remains effective as long as the commitment cost is set to a reasonable non-trivial value.

Table 10: Sensitivity analysis of the commitment cost $\beta$ on ISIC, SIIM, and BUSI datasets. The model shows robust performance across a range of standard values, with the default setting ($\beta = 0.25$) highlighted in **bold**.

| Commitment Cost ($\beta$) | ISIC (ACC / BMAC) | SIIM (ACC / BMAC) | BUSI (ACC / BMAC) |
|---|---|---|---|
| 0.10 | 89.94 / 86.05 | 81.85 / 81.90 | 93.25 / 95.15 |
| **0.25 (Default)** | **90.11 / 86.22** | **82.01 / 82.01** | **93.59 / 95.38** |
| 0.50 | 90.02 / 86.14 | 81.92 / 81.95 | 93.41 / 95.22 |
| 1.00 | 89.85 / 85.92 | 81.70 / 81.65 | 93.10 / 94.95 |

