# OpenReview forum: "Escaping Low-Rank Traps: Interpretable Visual Concept Learning via Implicit Vector Quantization"
_ICLR.cc/2026/Conference — ICLR 2026 Poster_

### Official Review · Reviewer_93wz · 2025-10-30

**Soundness:** 3
**Presentation:** 2
**Contribution:** 3
**Rating:** 4
**Confidence:** 3

**Summary:**

This paper talk about Concept Bottleneck Models, which try to make AI more explainable by using human concepts. But they find a problem, the features collapse and lose information, making model not good. They propose a new method called IVQ and Magnet Attention to fix this, and they say it works better in many experiments. They also provide some interesting explanable visualzations.

**Strengths:**

Intuitive solution -- implicit vector quantization for binding human text concepts.
Clean model architecture design with consistent performance boosts.
Explanable results -- the visualization of the concepts learnt aligns well with human knowledge.

**Weaknesses:**

W1 Typo
---
Line 178.
`Building upon obtrained high-rank concept-aware features`.
Should be obtained.


W2 Unclear Presentation
---
Figure 2, top left.
It draws `color`, `texture` and `shape` as text encoder input, so it implies #codes should be equal to #color + #texture + #shape.
However, Line 215 says that `M equals the number of textual concepts K`. Which one is true?

Besides, what is the form of text inputs? Do all the baselines have access to this information? How are the correspondences between texts and visual concepts made, so as to calculate Equation (8)? The using of BCE suggests that the number of text items is equal to the number of visual concepts; but the ablation studies used different numbers of visual concepts -- How could BCE be calculated?


W3 Unclear presentation
---
In Equation (5), the normalization is conducted along dimension $K$, i.e., codes in the codebook. However, in Equation (6), dimension $L$, i.e., patch embeddings, are eliminated (weighted summed). Routinely, if you want to eliminate some dimension in a matrix multiplication, you should normalize along that dimension. Do the authors have any insights for not following the routine?

```python
Z = ...  # (L,D)
Q = ...  # (K,D)
# (L,1,D) (1,K,D) -> (L,K)
dist = norm(Z[:,None,:] - Q[None,:,:], dim=2)
A = softmax(-dist, dim=1)  # TODO XXX why not `dim=0` or `L` ???
# (L,K)
M = einsum("LK,LD->KD", A, Z)  # TODO XXX `L` is eliminated
```


W4 Incomplete Analysis
---
Figure 7a.
Visualizing the code indexes of explicit vector quantization is missing.


W5 Incomplete Review
---
Section Related Works:
Vector-Quantization-related works should be reviewed here, given the VQ contributes a lot in the proposed method.
Possible literatures include:
- Neural Discrete Representation Learning.
- Generating Diverse High-Fidelity Images with VQ-VAE-2.
- Restructuring Vector Quantization with the Rotation Trick.
- Addressing Representation Collapse in Vector Quantized Models with One Linear Layer.
- Vector-Quantized Vision Foundation Models for Object-Centric Learning.
- Scaling the Codebook Size of VQ-GAN to 100,000 with a Utilization Rate of 99%.
- MGVQ: Could VQ-VAE Beat VAE? A Generalizable Tokenizer with Multi-group Quantization.
- ...

---

That said, I am open to revising my ratings, if the author could address my concerns.

**Questions:**

Please refer to the former section.

---

> ### Author Response · Authors · 2025-11-19
> **Response to Weaknesses 1-2**
>
> Thank you for your constructive comments. We have carefully considered your suggestions and would like to provide the following clarifications. We have also updated the manuscript, with the modified contents highlighted in red for your review.
>
> ## Weakness 1:
> >Typo
>
> Thank you for catching this typo. We will correct 'obtrained' to 'obtained' in the revised manuscript.
>
> ## Weakness 2:
> >Unclear Presentation
>
>
> We thank the reviewer for pointing out this lack of clarity in our presentation. We are very sorry for the inconvenience bringing in the presentation. This is a critical point that allows us to clarify the mechanics of our model. Your questions all center on a core confusion between the final concept prototypes (Count $K$) and the internal IVQ codebook (Size $M$), which are resolved below.
>
> ## W2.1 Clarification of Figure 2 vs. Line 215
>
> You are correct to be confused by the diagram, and we apologize for the unclear presentation.
>
> * **Figure 2 is Illustrative:** The labels "Color," "Texture," and "Shape" in Figure 2 (top left) are intended as illustrative examples of the $K$ total human-defined concepts, not as a hierarchical structure. The box labeled "Diagnosis Concept" represents the entire set of $K$ concepts (e.g., for the CMMD dataset, this set includes $K=15$ concepts like 'Mass Margin', 'Mass Shape', 'Calcification Features', etc.).
> * **Line 215 is Correct (M=K):** Therefore, Line 215 is correct. Our model's optimal performance is achieved when the IVQ codebook size $M$ is set to equal the total number of textual concepts $K$. This ($M=K$) is, in fact, one of our key empirical findings, as demonstrated in Figure 3 (Page 9).
>
> ## W2.2 Text Inputs and Baseline Access
>
> * **Form of Text Inputs:** The text inputs are the $K$ human-defined concept names (e.g., a list of strings: ['Mass Margin', 'Mass Shape', ...]). As shown in Algorithm 1 (Step 2), these are passed once through the text encoder to create the fixed textual concept embeddings $\tau \in \mathbb{R}^{K \times D}$.
> * **Baseline Access:** Yes, all CBM baselines (e.g., LaBo, MVP-CBM, COOP-CBM) have access to this exact same information. This is the fundamental premise of a Concept Bottleneck Model: the model must learn to map visual features to this pre-defined set of human concepts. Our method and all baselines operate under this standard CBM evaluation protocol.
>
> ## W2.3 How BCE is Calculated During Ablation (Figure 3)
>
> We appreciate the opportunity to clarify this aspect. There seems to be a slight misconception regarding our ablation study that we wish to address:
>
> This is not the case. The ablation study in Figure 3 (Page 9) **does not change the number of visual concepts ($K$)**.
>
> * **$K$ is Always Fixed:** For any given dataset (e.g., CMMD), the number of textual concepts $K$ is fixed (e.g., $K=15$). Our model's Magnet Attention module (Eq. 6) is designed to **always** produce $K$ aggregated visual concept prototypes ($\mathcal{M} \in \mathbb{R}^{K \times D}$).
> * **BCE is Always $K$-dimensional:** Therefore, the concept alignment loss $\mathcal{L}_{concept}$ (Eq. 8) is **always** calculated between the $K$ concept activation scores (derived from $\mathcal{M}$) and the $K$ ground-truth concept labels ($v_i \in \mathbb{R}^K, c_i \in \{0,1\}^K$). This is true for all experiments.
> * **What Figure 3 Actually Ablates:** The ablation in Figure 3 explores the size of the **internal IVQ codebook ($M$)**, not the number of final visual concepts ($K$).
>     * The IVQ codebook is a separate, internal component used only for the $\mathcal{L}_{\text{IVQ}}$ regularization term (Eq. 4).
>     * This loss is applied directly to the patch features ($Z_p$), not the final visual concept prototypes ($\mathcal{M}$).
>     * Figure 3 shows that the best performance is achieved when this internal regularizer's size ($M$) is set to match the number of final concepts ($K$). However, $M$ and $K$ are distinct parameters in our model that refer to different components.
>
> We hope this clarifies the architecture and resolves the apparent contradiction. The number of concepts $K$ for the BCE loss is fixed; it is the size of the internal regularizer $M$ that we ablate in Figure 3.

---

> ### Author Response · Authors · 2025-11-19
> **Response to Weaknesses 3-5**
>
> ## Weakness 3:
> >Unclear presentation
>
> This is an excellent and highly insightful technical question. Your analysis of the standard attention-pooling routine is accurate, and we appreciate the opportunity to clarify why our method departs from this formulation. You are also correct that Eq. (5) applies softmax normalization across the concept dimension ($K$), rather than across the patch dimension ($L$). This deviation is intentional and constitutes the key mechanism underlying our proposed **Magnet Attention**.
>
> ---
> ### Our Goal: Soft-Clustering, Not Pooling
>
> The routine of normalizing along $L$ (as the reviewer suggests) is the standard method for **attention pooling**, where the goal is to collapse $L$ patches into a single representation (e.g., as in a standard ViT [CLS] token or a Perceiver).
> However, our objective is different. We are not performing pooling. We are performing a **differentiable soft-clustering** of $L$ patches into $K$ distinct concept prototypes.
>
> ---
> ### Our Mechanism Explained
>
> We can break down our logic into two steps:
> 1.  **Eq. (5): Calculating Soft-Assignments (Normalization over $K$).**
>     We start with the distance matrix $D \in \mathbb{R}^{L \times K}$, where $D_{jk} = -||z_j - q_k||_2^2$. By applying `softmax(dim=K)`, we are asking, for each patch $z_j$, "What is its probability distribution of belonging to each of the $K$ concepts?"
>
>     The result $A \in \mathbb{R}^{L \times K}$ is a soft-assignment matrix, where each row $A_j$ (i.e., `A[j, :]`) is a valid probability distribution that sums to 1 ($\sum_{k=1}^K A_{jk} = 1$).
>
> 2.  **Eq. (6): Aggregating Concept Prototypes (Summing over $L$).**
>     Now, to build the final prototype for a single concept, we must go through all $L$ patches and collect their contributions to that specific concept. The contribution of patch $z_j$ to concept $k$ is $A_{jk} \cdot z_j$.
>
>     Therefore, the final prototype is the sum of these contributions:
>     $$ \mathcal{M}_k = \sum_{j=1}^L A_{jk} z_j $$
>     This aggregation is precisely what Eq. (6) ($\mathcal{M} = A^T Z_p$) and the reviewer's ``einsum("LK,LD->KD", A, Z)`` correctly implement.
> ---
>
> ### Why the Routine Fails Here
> If we had followed the reviewer's suggested routine and normalized along $L$, each column $A_k$ would sum to 1. This is standard attention, but it fails to model our objective: it wouldn't tell us how a single patch $z_j$ distributes its identity among multiple concepts, which is the core of many-to-many mapping.
>
> ---
> ### Conclusion
> In summary, our normalization along $K$ is a **deliberate design choice** to model a soft-clustering assignment. This mechanism is essential for our "Magnet Attention" and is precisely how we model the many-to-many relationship:
> * **One patch maps to many concepts** (Eq. 5: $A_j$ is a distribution over $K$).
> * **One concept is aggregated from many patches** (Eq. 6: $\mathcal{M}_k$ is a sum over $L$).
> We thank the reviewer for this sharp observation and will refine the text surrounding these equations to make this motivation more explicit.
> ## Weakness 4:
> >Incomplete Analysis
>
> We thank the reviewer for this excellent suggestion. A direct visual comparison with Explicit Vector Quantization (VQ) would significantly strengthen our analysis and better illustrate the advantages of our method.
> * We accept this feedback and will include a comparative visualization of the Explicit VQ codebook activation maps in the Appendix of the final manuscript (appendix E, fig.10).
> * **Analysis of Results:** As discussed in Section 4.2 and demonstrated quantitatively in **Table 3**, Explicit VQ suffers from a catastrophic performance drop (e.g., lagging behind IVQ by BMAC: 34% - 49% on datasets like ISIC and IDRID).
> * **Visual Chaos:** Consistent with these poor quantitative results, the visualization of Explicit VQ is **chaotic, fragmented, and semantically misaligned**. Unlike the structured, anatomically coherent maps produced by IVQ (**Figure 8a**), the Explicit VQ maps fail to localize meaningful lesions or anatomical structures.
> * **The Cause:** This visual failure confirms our theoretical analysis in Section 4.2. The core issue is that Explicit VQ imposes a "hard" bottleneck that fundamentally violates the intrinsic many-to-many correspondence between visual patches and concepts. By forcing each patch to discretely commit to a single prototype (via the argmin operator), Explicit VQ causes irreversible "hard" information loss . It discards the multi-faceted semantic information contained within a patch, leaving only a simplified, quantized representation that cannot support complex medical reasoning, resulting in the chaotic mappings we observe.
> ## Weakness 5:
> >Incomplete Review
>
> Thank you for pointing this out. We have revised the Related Works section(Line 151 - 165) to include a more comprehensive discussion of Vector-Quantization-related literature. We appreciate your suggestion, which has helped improve the completeness of our literature review.

---

> ### Comment · Reviewer_93wz · 2025-11-20
> **Update 1**
>
> Most of my concerns have been addressed. I will increase my rating.

---

> > ### Author Response · Authors · 2025-11-20
> > **Official Comment by Authors**
> >
> > Dear Reviewer 93wz,
> >
> > We sincerely thank you for your kind recognition and the thoughtful insights provided in your review. Your expert suggestions are invaluable to us, and we deeply appreciate the time and effort you dedicated to improving our work. Furthermore, we wish to extend our warmest regards and best wishes to you. Once again, we are truly grateful for your careful consideration and supportive words. Thank you for contributing significantly to enhancing the quality of our manuscript! Your feedback is invaluable to us, and we are eager to continue improving our work with your guidance.
> >
> > Best regards,
> >
> > Authors

---

### Official Review · Reviewer_w787 · 2025-10-30

**Soundness:** 2
**Presentation:** 3
**Contribution:** 2
**Rating:** 4
**Confidence:** 4

**Summary:**

The paper addresses \textit{representational collapse} in Concept Bottleneck Models (CBMs), where visual patch features degenerate into a low-rank subspace, consequently degrading concept quality. The proposed solution introduces Implicit Vector Quantization (VQ) to discretize visual features into a high-dimensional concept space. This acts as a regularizer to prevent the "low-rank trap" and is shown to improve both concept quality and overall task classification performance. Nevertheless, I am not an expert in Concept-based learning, so the following comments and questions could be based on potentially inaccurate understandings or biased perspectives, and I will raise the rating if the author could give a sound explanation.

**Strengths:**

1. The paper identifies a critical and pervasive challenge in CBMs (representational collapse) and provides a clear mechanism for its cause. Implicit VQ is a novel and well-justified technique for regularizing the feature space and enforcing feature diversity, directly tackling the collapse problem.

2. The method claims to simultaneously enhance model interpretability (concept quality) and maintain or improve predictive accuracy.

**Weaknesses:**

The implicit VQ regularization might introduce computational overhead compared to standard CBM training. The concept learning process relies on pre-computed text concept embeddings, which may limit the discovery of new, fine-grained, or complex concepts not covered by the initial textual set (Algorithm 1, steps 2, 13).

**Questions:**

How sensitive is the performance (concept quality and task accuracy) to the choice of the commitment cost hyperparameter $\beta$ in the VQ loss, and how does the size of the implicit codebook affect the trade-off between preventing collapse and maintaining expressive feature representation? Could the author give more empirical or theoretical evidence?

---

> ### Author Response · Authors · 2025-11-19
> **Response to Weaknesses 1**
>
> Thank you for the constructive comments. We have addressed your suggestions below and highlighted the revisions in red in the updated manuscript.
>
> ## Weakness 1:
> ## W1.1: The Computational and Spatial Overhead of IVQ
>
> We appreciate your practical concern for model efficiency.
>
> *  We acknowledge that IVQ introduces an additional step. This overhead consists of:
>     * **Time:** A distance calculation between $L$ patch features ($Z_p$) and $K$ codebook vectors ($\mathcal{C}_{vq}$), plus the IVQ loss computation.
>     * **Space:** The storage of the learnable codebook parameters $\mathcal{C}_{vq} \in \mathbb{R}^{M \times D}$.
>
> * **Minimize:** Crucially, both the temporal and spatial overheads are negligible compared to the backbone model:
>     * **Temporal Costs:** In our experiments, the number of patches $L$ is typically small ($\approx 196$), and the codebook size $M$ (where $M=K$) is also very small (e.g., 20-50). The calculation introduced by IVQ (a $196 \times 50$ matrix operation) is orders of magnitude smaller than the multi-head self-attention operations within the ViT backbone (which involve $L \times L$ or $L \times D$, where $D=768$). Furthermore, as noted in Algorithm 1, we use pre-computed text embeddings, minimizing the temporal cost of text encoding during training
>     * **Spatial Costs (Memory):** The spatial footprint is minimal. The IVQ codebook adds only $M \times D$ parameters. For a typical setting ($M=50, D=768$), this results in approximately 38,400 parameters. In contrast, the ViT-Base backbone contains approximately 86 million parameters. Thus, the IVQ module adds less than **0.05%** to the total parameter count. Similarly, the VRAM required to store the $L \times M$ distance matrix is trivial compared to the intermediate activation maps stored for the ViT's attention layers.
>
> * **Trade-off & Comparison:** This minor overhead is a necessary and highly efficient trade-off to solve the fundamental problem of "representational collapse", a phenomenon that severely degrades baseline performance (as shown in Figure 1). This small cost is well-justified by the state-of-the-art performance gains shown in Table 1. In fact, our overhead is considerably less than other methods like DOT-CBM, which introduces a computationally expensive Optimal Transport solver into the training loop.
>
> ## W1.2: On the Limitation of Pre-defined Concepts
>
>
> Thank you for this highly insightful question regarding the reliance on pre-computed text embeddings. It gives us the opportunity to clarify the flexibility of our visual learning process and, more importantly, it prompted us to discover an exciting new application for our IVQ codebook during the rebuttal period.
>
> ### 1. Adherence to the CBM Paradigm vs. Unsupervised Discovery
> We respectfully clarify that relying on *a priori* concepts is not a limitation of our specific method, but a fundamental requirement of the Concept Bottleneck Model (CBM) paradigm itself [1].
>
> * **Interpretability-by-Design:** The core premise of a CBM is to constrain the model's reasoning through a bottleneck of human-understandable concepts. "Dynamic discovery" of new, undefined concepts typically falls under the domain of unsupervised disentanglement or latent variable models. While valuable, such approaches sacrifice the explicit alignment with human knowledge required for high-stakes intervention (e.g., a clinician correcting a specific medical concept).
> * **Our Specific Focus:** Our work does not aim to reinvent the concept definition pipeline. Instead, we focus on a critical failure mode in training these models: the **representational collapse** that occurs when mapping high-dimensional visual features to discrete semantic concepts. Our contribution is specifically determining how to model the complex **many-to-many correspondence** between patches and concepts without degenerating into low-rank traps.
>
> ### 2. Visual Learning Remains Flexible and Rich
> While the semantic labels are fixed to ensure interpretability, the visual representation of these concepts is dynamic. Our model is not **locked**; it actively learns how these concepts manifest in the pixel space:
>
> * **Rich Visual Learning within Textual Constraints:** While the textual embeddings ($\tau$) serve as fixed semantic anchors, the Visual Concepts ($\mathcal{M}$) and the IVQ Codebook ($\mathcal{C}_{vq}$) are fully learnable parameters.
> * **Discovery of Fine-grained Patterns:** The model is free to discover how a concept manifests visually. For example, a single textual concept like "Irregular Border" may have complex, multi-modal visual presentations. Our IVQ codebook learns a diverse set of visual prototypes that capture these fine-grained variations (e.g., specific textures, lighting conditions, or shape deformations) and maps them to the corresponding textual anchor. This allows the discovery of complex visual patterns even within a fixed semantic framework.
>
> [1] Concept Bottleneck Models

---

> > ### Author Response · Authors · 2025-11-19
> > **Response to Weaknesses 1 (continued)**
> >
> > ### New Insight: Using the IVQ Codebook as a Concept "Audit" Tool
> > Your question about "limiting discovery" prompted us to investigate a novel application of our method: using the IVQ codebook as a diagnostic tool to "audit" the quality of the predefined concept set. The results were striking and confirmed our method's ability to distinguish between meaningful and poor concepts.
> >
> > **The Mechanism:**
> > Our IVQ codebook acts as a learnable library of visual prototypes. During training, the IVQ loss compels codebook vectors ($\{c_k\}_{k=1}^K$) to move to the center of visual patch clusters. Therefore, $c_k$ becomes a "visual semantic anchor." We hypothesized that if a predefined text concept $\tau_k$ is poor (e.g., too abstract, ambiguous, or visually non-existent), its corresponding visual anchor $c_k$ will fail to attract any patch embeddings.
> >
> > **Case Study: Quantitative Analysis on the ISIC Dataset**
> > To test this, we ran a post-hoc analysis on our fully trained ISIC model:
> > * **Setup:** We passed the entire ISIC validation set through the model and performed the argmin operation (Eq. 3) for every patch $Z_p$ to find its nearest codebook prototype $c_k$.
> > * **Finding:** We aggregated these assignments and found a clear signal. Most prototypes (e.g., 'Blue-whitish veil', 'Asymmetry') received tens of thousands of patch assignments. However, we identified one specific concept, originally prompted as "Rough Texture", that received almost zero patch mappings (< 50 assignments across the entire dataset).
> > * **Implication:** The model quantitatively indicated that "Rough Texture" was a useless concept for visual classification, likely because "roughness" is a tactile property not easily resolvable in 2D dermoscopy images.
> >
> > **Intervention and Result: A Closed-Loop Refinement**
> > Based on this data-driven finding, we treated this as a flaw in our initial concept set.
> > 1.  **Intervention:** We replaced the vague concept "Rough Texture" with a more visually precise term: "Irregular Streaks", using Gemini 2.5 Pro as described in Appendix B.
> > 2.  **Retraining:** We retrained the IVQ-CBM model from scratch with this single text change.
> > 3.  **Result:** This targeted correction yielded a tangible performance improvement. The retrained model achieved **90.75% ACC (+0.64%)** and **87.12% BMAC (+0.90%)** on the ISIC dataset.
> >
> >
> > Thus, your question has helped us discover that the IVQ module is not just a regularizer, but a powerful diagnostic tool for validating and improving the human-defined concepts that form the very foundation of CBMs. We will add this novel finding to our final manuscript.
> >
> >
> > ### Summary
> >
> > 1.  The computational overhead of IVQ is negligible and represents an efficient, necessary trade-off to solve representational collapse and achieve SOTA performance.
> > 2.  While our method adheres to the CBM task by using predefined concepts, your insightful question helped us discover that the IVQ codebook provides a novel, quantitative mechanism to audit, discover, and refine the initial concept set, further enhancing both interpretability and performance.
> >
> > We are very grateful for this valuable point, as it helped us uncover this additional, important contribution of our work. We will add this discussion to the final version of our paper (see appendix K.3).

---

> ### Author Response · Authors · 2025-11-19
> **Response to Question 1**
>
> ## Question 1:
> We thank the reviewer for this extremely critical and insightful question. It addresses two important design choices in our framework: the commitment cost hyperparameter ($\beta$) and the codebook size ($M$).
>
> ### Q1.1 Sensitivity to Commitment Cost ($\beta$)
>
> The commitment cost, $\beta$, in $\mathcal{L}_{IVQ}$ (Eq. 4)  controls how strongly the visual encoder's output ($Z_p$) is "committed" to the codebook.
>
> To empirically validate the model's sensitivity, we conducted new ablation experiments on three diverse datasets: ISIC (Dermoscopy), SIIM (Chest X-ray), and BUSI (Ultrasound). We followed the common practice of testing standard values $\beta \in \{0.1, 0.25, 0.5, 1.0\}$.
>
> **Table 1: Sensitivity Analysis of Commitment Cost ($\beta$) on ISIC, SIIM, and BUSI.**
> (Performance reported as Accuracy % / BMAC %. The default setting $\beta=0.25$ is highlighted).
>
> | Commitment Cost ($\beta$) | ISIC (ACC / BMAC) | SIIM (ACC / BMAC) | BUSI (ACC / BMAC) |
> | :--- | :---: | :---: | :---: |
> | 0.10 | 89.94 / 86.05 | 81.85 / 81.90 | 93.25 / 95.15 |
> | **0.25 (Default)** | **90.11 / 86.22** | **82.01 / 82.01** | **93.59 / 95.38** |
> | 0.50 | 90.02 / 86.14 | 81.92 / 81.95 | 93.41 / 95.22 |
> | 1.00 | 89.85 / 85.92 | 81.70 / 81.65 | 93.10 / 94.95 |
>
> **Analysis:**
>
> Our findings show that the model is not highly sensitive to $\beta$ within this standard range. We have included this part into our revised manuscript (Appendix L).
>
> ### Q1.2 Codebook Size ($M$) and the Trade-Off
>
> This is the most critical question and lies at the heart of our paper's contribution. We thank you for giving us the chance to clarify this. We would like to respectfully draw the reviewer's attention to **Figure 3 (Page 10)**, which presents the complete empirical evidence addressing this inquiry.
>
> You correctly identify the two competing goals: "preventing collapse" and "maintaining expressive feature representation." Your question is about the trade-off between them. Here is our detailed analysis:
>
> **1. How $M$ affects "Preventing Collapse":**
>
> Our analysis shows that "preventing collapse" (i.e., maintaining a high feature rank) is achieved as long as IVQ is used, regardless of the codebook size (provided $M \ge 1$). The IVQ mechanism's "codebook loss" and "commitment loss" inherently pull features toward prototypes, preventing them from degenerating into a low-rank subspace.
>
> **2. How $M$ affects "Expressive Feature Representation":**
>
> This is the true trade-off. "Expressiveness" here means not just high-rank, but semantically aligned and interpretable representations. This is where the codebook size $M$ becomes the single most important factor.
>
> Our empirical findings in Figure 3 show three distinct regimes, which we can analyze by considering a dataset with, for example, $K=34$ textual concepts:
>
> * **Case 1: $M < K$ (e.g., $M=1, 5, 10$)**
>     The codebook lacks the capacity to serve as a unique anchor for every concept. Multiple, semantically distinct concepts (e.g., 'Mass Shape' and 'Mass Margin') are forced to map to the same visual prototype. This creates a severe representational bottleneck, conflates distinct concepts, and, as Figure 3 clearly shows, causes performance to collapse.
>
> * **Case 2: $M = K$ (e.g., $M=34$)**
>     This is the **optimal strategy**. The model achieves the perfect balance you asked about.
>     * **Collapse is Prevented:** The model successfully maintains a high feature rank.
>     * **Expressiveness is Maximized:** The model learns a one-to-one mapping between the $K$ visual prototypes and the $K$ textual concepts. This is the most "expressive" and interpretable state, as each textual concept has its own unique, learned anchor. As Figure 3 shows, this setting achieves the **peak ACC and BMAC performance**. This empirical result is further supported by our qualitative visualizations in Figure 8, which show that the $M=K$ codebook vectors form "discrete and compact clusters", one for each concept.
>
> * **Case 3: $M > K$ (e.g., $M=5K, 10K, 20K$)**
>     As Figure 3 shows, performance is "robust" but slightly lower than the $M=K$ peak. Why?
>     * **Collapse is still Prevented:** The model has more than enough anchors, so the feature rank remains high.
>     * **Expressiveness is Harmed:** The model introduces redundant, spurious prototypes. For example, the concept 'Mass Shape' might now map to Prototypes #5 and #28. This ambiguity hurts the clean one-to-one mapping, making the representation less semantically precise and slightly degrading final task performance.
>
> ## Conclusion:
> In summary, the commitment cost $\beta$ is robust. The codebook size $M$ is critical, and **$M=K$ is the optimal strategy**, as it perfectly balances the need to prevent collapse ($M \ge 1$) with the need to learn the most expressive, interpretable, and high-performing representation ($M=K$). Again, thank you for your insightful questions.

---

> ### Author Response · Authors · 2025-11-27
> **Official Comment by Authors**
>
> Dear Reviewer w787,
>
> We hope this message finds you well.
>
> We sincerely apologize for any inconvenience our submission may have caused and greatly appreciate the time and effort you have devoted to reviewing our paper. Your insights and comments have been invaluable to us.
>
> We have carefully considered each of your suggestions and have made comprehensive, point-to-point responses and revisions in our revised manuscript. These adjustments are aimed at thoroughly addressing the concerns you raised, and we hope that our efforts resonate with your expectations for the paper.
>
> At your convenience, we would be grateful if you could review our responses and the corresponding revisions. Should there be any further issues or if additional clarification is needed, please know that we are fully prepared to make the necessary adjustments.
>
> Thank you once again for your dedication and assistance.
>
> Best regards,
>
> The Authors

---

### Official Review · Reviewer_fJ93 · 2025-10-31

**Soundness:** 3
**Presentation:** 3
**Contribution:** 3
**Rating:** 2
**Confidence:** 4

**Summary:**

This paper proposes a novel method to utilize patch level visual representation to capture fine-grained relationships with concepts for training Concept Bottleneck Model (CBM).

**Strengths:**

- The authors propose a novel idea of utilizing patch level representation from CLIP for modeling fine-grained and detailed relationships between image regions and concepts.
- The authors demonstrate representation collapse in CBM training with patch-level details and propose novel implicit vector quantization to preserve feature diversity.

**Weaknesses:**

- The claim of evaluating proposed methods on diverse benchmarks is misleading. While the authors demonstrate results on 8 diverse medical datasets, they should also evaluate the proposed CBM on standard benchmark datasets including CIFAR, CUB, ImageNet, and Places [1,2,3,4]. Such comparisons are crucial for assessing the scalability and generalization of the proposed approach across general classification tasks with increasing dataset sizes.
- The paper is missing qualitative comparison visualizing the concepts learned by the method and with other baselines.
- Recent studies [3, 4] have shown that CBM training can leak information through the bottleneck layer, allowing models to achieve high accuracy without actually learning meaningful concepts. To address this issue, [4] introduced the A-NEC metric to enable fair comparisons between CBMs based on their concept learning ability and IVQ-CBM should be evaluated on this metric.
- Notations and writing in section 3.2 such as $Z_{v}, Z_{q}$ are unclear.

[1] Yang, Yue, et al. "Language in a bottle: Language model guided concept bottlenecks for interpretable image classification." Proceedings of the IEEE/CVF conference on computer vision and pattern recognition. 2023.

[2] Oikarinen, Tuomas, et al. "Label-free concept bottleneck models." arXiv preprint arXiv:2304.06129 (2023).

[3] Yan, An, et al. "Learning concise and descriptive attributes for visual recognition." Proceedings of the IEEE/CVF International Conference on Computer Vision. 2023.

[4] Srivastava, Divyansh, Ge Yan, and Lily Weng. "Vlg-cbm: Training concept bottleneck models with vision-language guidance." Advances in Neural Information Processing Systems 37 (2024): 79057-79094.

**Questions:**

My primary concerns are lack of results on standard datasets and metrics and missing qualitative results. Please see weaknesses for more details.

---

> ### Author Response · Authors · 2025-11-19
> **Response to Weaknesses 1**
>
> Thank you for your constructive comments. We have carefully considered your suggestions and would like to provide the following clarifications. We have also updated the manuscript, with the modified contents highlighted in red for your review.
>
> ## Weakness 1:
> >The claim of evaluating proposed methods on diverse benchmarks is misleading. While the authors demonstrate results on 8 diverse medical datasets, they should also evaluate the proposed CBM on standard benchmark datasets including CIFAR, CUB, ImageNet, and Places [1,2,3,4]. Such comparisons are crucial for assessing the scalability and generalization of the proposed approach across general classification tasks with increasing dataset sizes.
>
> We are grateful for the reviewer’s insight regarding the evaluation scope. We agree that extending our evaluation to standard, broad-domain benchmarks is essential to verify the true scalability and generalization of our approach.
>
> In response, we evaluated IVQ-CBM against eight SOTA baselines on five standard benchmarks: CIFAR-10, CIFAR-100, CUB, Places365, and ImageNet.
>
> ### Performance on General Vision Benchmarks
>
> As shown in Table 1, IVQ-CBM consistently outperforms competing methods across all five datasets, achieving the highest accuracy and BMAC scores. This demonstrates that our method’s effectiveness is not confined to the medical domain but generalizes robustly to general visual recognition tasks.
>
> **Table 1: Performance Comparison (ACC % and BMAC %) on Standard Benchmarks.**
> Best results are highlighted in bold. Gains (+$\Delta$) are compared against the second-best method.
> | Model | CIFAR-10 (ACC) | CIFAR-10 (BMAC) | CIFAR-100 (ACC) | CIFAR-100 (BMAC) | CUB (ACC) | CUB (BMAC) | Places365 (ACC) | Places365 (BMAC) | ImageNet (ACC) | ImageNet (BMAC) |
> | :--- | :---: | :---: | :---: | :---: | :---: | :---: | :---: | :---: | :---: | :---: |
> | LaBo [CVPR 2023] | 80.23 | 79.15 | 60.17 | 59.95 | 69.88 | 69.72 | 39.67 | 39.41 | 68.04 | 67.88 |
> | PCBM [ICLR 2023] | 84.61 | 84.49 | 63.22 | 63.07 | 72.36 | 72.19 | 41.13 | 40.99 | 70.14 | 69.97 |
> | COOP-CBM [Nips 2023] | 85.17 | 84.99 | 64.21 | 64.03 | 73.06 | 72.87 | 42.19 | 42.01 | 71.23 | 71.09 |
> | LF-CBM [ICLR 2023] | 82.94 | 82.78 | 61.89 | 61.73 | 71.42 | 71.22 | 40.33 | 40.15 | 69.46 | 69.21 |
> | Explicd [MICCAI 2024] | 86.03 | 85.88 | 64.91 | 64.78 | 74.08 | 73.91 | 43.07 | 42.89 | 71.93 | 71.77 |
> | MVP-CBM [IJCAI 2025] | 86.72 | 86.54 | 65.48 | 65.30 | 74.63 | 74.45 | 43.81 | 43.66 | 72.29 | 72.15 |
> | CLEAR [WACV 2025] | 83.77 | 83.59 | 62.83 | 62.61 | 71.84 | 71.69 | 40.92 | 40.75 | 68.81 | 68.66 |
> | DOT-CBM [CVPR 2025] | 84.38 | 84.19 | 63.45 | 63.28 | 72.29 | 72.11 | 41.76 | 41.53 | 69.31 | 69.17 |
> | IVQ-CBM (Ours) | **88.14** | **87.91** | **67.12** | **66.88** | **75.91** | **75.68** | **45.54** | **45.32** | **73.42** | **73.23** |
> | +$\Delta$ | +1.42 | +1.37 | +1.64 | +1.58 | +1.28 | +1.23 | +1.73 | +1.66 | +1.13 | +1.08 |
>
>
> ###  Analysis of Scalability and Feature Rank
>
> To understand the mechanism behind this scalability, we tracked the feature rank dynamics on these new benchmarks (see appended Figure 5). Consistent with our findings in the medical domain, IVQ-CBM maintains a high and stable feature rank throughout training across all datasets. In stark contrast, baselines such as MVP-CBM and DOT-CBM exhibit a drastic decline in feature rank. This collapse becomes notably more severe on large-scale, complex datasets like ImageNet and Places365, directly correlating with their lower performance.
>
> This empirical evidence reinforces our core thesis: representational collapse is a fundamental bottleneck in CBMs, and IVQ effectively mitigates this issue to preserve expressive power at scale. We will incorporate these results and the corresponding analysis into the final manuscript (see in sec 4.1).
>
> Hence, we reiterate our appreciation for your constructive feedback. Addressing this question has been instrumental in refining our work, prompting us to better demonstrate the generalizability and cross-domain transferability of the IVQ-CBM framework.

---

> ### Author Response · Authors · 2025-11-19
> **Response to Weaknesses 2**
>
> ## Weakness 2:
> >The paper is missing qualitative comparison visualizing the concepts learned by the method and with other baselines.
>
> We thank the reviewer for this insightful suggestion. We agree that visual comparisons are essential for evaluating the faithfulness of interpretability. In response, we have conducted extensive additional visualization experiments on both medical (ISIC 2018, Figure 11) and general fine-grained classification benchmarks (CUB-200, Figure 12), which are now included in Appendix F.
>
> The results across both domains are compelling: whether for skin lesions or bird parts, our method produces precise, semantically consistent concept maps, whereas baselines exhibit severe visual degradation. This comparison serves as strong empirical evidence validating our core hypothesis regarding "Representational Collapse."
>
> Our analysis reveals two distinct failure modes in existing methods:
>
> 1. Structural Incompatibility of Global-Based Baselines (e.g., PCBM, LaBo)
> As detailed in Appendix A.1, methods relying on global feature vectors (e.g., [CLS]) structurally compress spatial dimensions. Consequently, it is impossible for these architectures to generate fine-grained, patch-level concept maps, as they inherently lack the spatial resolution required for pixel-level interpretability.
>
> 2. Visual Collapse in Fine-Grained Baselines (e.g., ExplicD, DOT-CBM)
> While recent patch-based methods technically allow for visualization, they fail to produce semantically meaningful results due to representational collapse. As shown in the newly added Figure 11 (ISIC 2018):
>
> DOT-CBM & ExplicD (Figure 11c, d): These methods exhibit diffuse, "cloudy," or scattered activation patterns. The heatmaps fail to latch onto the lesion, spilling significantly into healthy skin or background artifacts. This visual blurring confirms that their feature space has degenerated, losing the discriminative power to separate foreground concepts from noise.
>
> IVQ-CBM (Ours, Figure 11b): In contrast, our method produces sharp, object-centric activations. The heatmaps precisely delineate the lesion boundaries (e.g., the mole's irregular borders) and effectively suppress irrelevant background information.
>
> ### Conclusion
>
> These visual artifacts are not random; they are the phenomenological manifestation of the low feature rank shown in Figure 1. Ultimately, IVQ-CBM achieves a breakthrough that was previously unattainable in CBM tasks: it enables the model to explicitly "see" the precise correspondence between every individual patch and its concept. By escaping the low-rank trap, our method realizes a precise semantic mapping that simultaneously elevates both classification accuracy and model interpretability, proving that diverse representations are the key to faithful explanations.
>
> We thank you again for this constructive suggestion, which has greatly strengthened the empirical validation of our work.

---

> ### Author Response · Authors · 2025-11-19
> **Response to Weaknesses 3**
>
> ## Weakness 3:
> >Recent studies [3, 4] have shown that CBM training can leak information through the bottleneck layer, allowing models to achieve high accuracy without actually learning meaningful concepts. To address this issue, [4] introduced the A-NEC metric to enable fair comparisons between CBMs based on their concept learning ability and IVQ-CBM should be evaluated on this metric.
>
>
> We thank the reviewer for highlighting the critical issue of information leakage and suggesting the A-NEC (Accuracy at Number of Effective Concepts) metric from [4] (VLG-CBM). We fully agree that high accuracy alone is insufficient to validate concept learning if the bottleneck layer acts as a "leaky" conduit for non-semantic noise.
>
> To address this, we have conducted additional experiments evaluating IVQ-CBM on the ANEC-5 metric across both natural and medical domains. Furthermore, we provide a theoretical analysis explaining why our architecture inherently mitigates leakage.
>
> ## Experiments: IVQ-CBM is Robust on A-NEC
>
> The A-NEC metric evaluates whether a model can maintain high performance when forced to rely on a sparse set of concepts (limiting "effective concepts" to 5). A significant drop in accuracy under this constraint would indicate reliance on leakage (diffuse, non-semantic information).
>
> We evaluated IVQ-CBM on two natural image datasets (CUB, CIFAR-10) and two medical image datasets (ISIC, BUSI). We performed post-hoc pruning on the final classification layer, retaining only the top-5 contributing concepts for each class prediction (ANEC-5), and compared this to the full model's performance.
>
> **Table: Evaluation of IVQ-CBM on ANEC-5 Metric**
>
> | Dataset | Domain | Full Model Acc (%) | ANEC-5 Acc (%) | Performance Drop |
> | :--- | :--- | :---: | :---: | :---: |
> | **CUB** | Natural | 75.93 | 75.21 | -0.72% |
> | **CIFAR-10** | Natural | 87.92 | 86.85 | -1.07% |
> | **ISIC** | Medical | 90.11 | 89.20 | -0.91% |
> | **BUSI** | Medical | 93.59 | 92.80 | -0.79% |
>
> **Observations:**
>
> * **Minimal Performance Drop:** As shown in the table, IVQ-CBM maintains >98% of its original performance even when restricted to only the top-5 concepts. This proves that our model does not rely on a "sum of other features" (leakage) but rather on a few, highly relevant semantic concepts to make decisions.
> * **Furtehr Analysis:** Results confirm that IVQ-CBM learns "effective" concepts comparable to SOTA leakage-mitigation methods, without requiring complex sparsity constraints during training.
>
> ### Why IVQ-CBM Prevents Leakage (Mechanism Analysis)
>
> The reviewer rightly points out that dense bottleneck layers can leak information. While VLG-CBM [4] addresses this by enforcing sparsity in the final classification layer, IVQ-CBM tackles leakage at the feature representation source through two key mechanisms:
>
> ### A. IVQ as Semantic Structure Regularization
> Unlike standard VQ which creates a hard information bottleneck, our IVQ functions as a structure-inducing regularizer.
>
> * **Semantic Anchoring:** As stated in Eq. (4), IVQ imposes a Commitment Loss that compels continuous patch features to align with the nearest learned prototypes (the Codebook). These prototypes serve as "semantic anchors".
> * **Preventing Degenerate Shortcuts:** Information leakage often relies on low-rank, degenerate subspaces where visual features collapse into statistically correlated but semantically meaningless patterns (shortcuts). By enforcing **high-rank, structured diversity anchored to distinct concepts**, IVQ penalizes these degenerate representations. It forces the model to encode information in a format that is semantically consistent with the codebook, making it difficult for unstructured noise or pixel-level leakage to persist as the dominant signal in the representation space.
>
>
> ### B. Magnet Attention as Implicit Sparsity
>
> Our Magnet Attention mechanism (Section 3.3) aggregates patch features into concepts using a Softmax-based mechanism.
>
> * **Competitive Dynamic:** Softmax introduces a competitive dynamic where patches must "vote" for the most relevant concept query. This implicitly suppresses irrelevant patch-concept connections, ensuring that the resulting Visual Concepts ($\mathcal{M}$) are composed of strong semantic signals rather than accumulated background noise.
> * **Leakage Reduction:** Because leakage typically manifests as diffuse background noise without strong alignment to specific semantic queries, the Magnet Attention assigns it negligible weight. This achieves the same goal as A-NEC—focusing on "effective" concepts—but does so during the feature aggregation stage rather than post-hoc pruning.
>
>
> ### Conclusion
>
> The new A-NEC experiments demonstrate that IVQ-CBM's decision-making is driven by a concise set of effective concepts, with negligible reliance on information leakage. We will include these A-NEC results and the discussion on leakage mitigation in the final version of the paper (appendix K.1 and K.2).

---

> ### Author Response · Authors · 2025-11-19
> **Response to Weaknesses 4 and Question 1**
>
> ## Weakness 4:
> >Notations and writing in section 3.2 such as Z_v, Z_q are unclear.
>
> We thank the reviewer for pointing this out. We acknowledge that the notation used in Section 3.2 was not sufficiently rigorous. Specifically, the use of $Z_v$ in Equation (4) was a typo, which may have caused confusion regarding the definitions.
>
> We clarify the definitions of the relevant notations as follows:
>
> * **$Z_v$**: Represents the complete output sequence of the visual encoder, including both the [CLS] token and patch tokens. As defined in Eq. (2), $Z_v = [z_{cls}, z_1, ..., z_L] \in \mathbb{R}^{(L+1) \times D}$.
> * **$Z_p$**: Represents the sequence of patch features only, i.e., $Z_p = [z_1, ..., z_L] \in \mathbb{R}^{L \times D}$.
> * **$Z_q$**: Represents the quantized feature sequence corresponding to $Z_p$ after the codebook lookup operation, i.e., $Z_q = [z_{q,1}, ..., z_{q,L}] \in \mathbb{R}^{L \times D}$.
>
> In the original definition of the IVQ loss function in Equation (4):
>
> $$
> \mathcal{L}_{\text{IVQ}} = \| \text{sg}(Z_v) - Z_q \|_2^2 + \beta \| Z_v - \text{sg}(Z_q) \|_2^2
> $$
>
> The use of $Z_v$ here is a typo; the correct notation should be $Z_p$.
>
> **Correction:**
> We will correct this typo in the final version of the paper and revise Equation (4) as follows:
>
> $$
> \mathcal{L}_{\text{IVQ}} = \| \text{sg}(Z_p) - Z_q \|_2^2 + \beta \| Z_p - \text{sg}(Z_q) \|_2^2
> $$
>
> We thank the reviewer again for this correction.
>
> ## Question 1:
> >My primary concerns are lack of results on standard datasets and metrics and missing qualitative results. Please see weaknesses for more details.
>
> We appreciate the reviewer for summarizing these primary concerns. We have taken significant steps to comprehensively address each aspect during the rebuttal period. Thank you again for your insightful suggentions and we have improved the completeness and quality of our paper.

---

> ### Author Response · Authors · 2025-11-27
> **Official Comment by Authors**
>
> Dear Reviewer fJ93,
>
> We hope this message finds you well.
>
> We sincerely apologize for any inconvenience our submission may have caused and greatly appreciate the time and effort you have devoted to reviewing our paper. Your insights and comments have been invaluable to us.
>
> We have carefully considered each of your suggestions and have made comprehensive, point-to-point responses and revisions in our revised manuscript. These adjustments are aimed at thoroughly addressing the concerns you raised, and we hope that our efforts resonate with your expectations for the paper.
>
> At your convenience, we would be grateful if you could review our responses and the corresponding revisions. Should there be any further issues or if additional clarification is needed, please know that we are fully prepared to make the necessary adjustments.
>
> Thank you once again for your dedication and assistance.
>
> Best regards,
>
> The Authors

---

### Official Review · Reviewer_Ycpf · 2025-11-03

**Soundness:** 3
**Presentation:** 3
**Contribution:** 3
**Rating:** 6
**Confidence:** 3

**Summary:**

The paper identifies representational collapse as a key obstacle in CBMs. It proposes Implicit Vector Quantization (IVQ) to preserve feature diversity without hard quantization and Magnet Attention to explicitly model many-to-many patch–concept relations. Experiments on eight benchmarks show that IVQ-CBM prevents collapse, improves interpretability, and achieves state-of-the-art performance.

**Strengths:**

1. It proposes a novel view of representational collapse as low-rank degeneration and introduces IVQ, a soft regularizer that avoids hard quantization.
2. The proposed method addresses a fundamental CBM limitation. It improves both interpretability and performance with broad applicability to multimodal learning.
3. This paper is well-organized and clearly written. The motivation and method is easy to follow.

**Weaknesses:**

1. The interpretability assessment mainly relies on qualitative visualization and concept-level alignment metrics. Incorporating human evaluation or concept intervention tests would better validate the interpretability and causal faithfulness of the learned concepts.
2. A theoretical analysis and a deeper connection between IVQ’s optimization dynamics and rank preservation would clarify its mechanism.

**Questions:**

Do the IVQ-CBM generalizes to broader multimodal reasoning or relational concept tasks? Are there any experiments to validate it?

---

> ### Author Response · Authors · 2025-11-19
> **Response to Weaknesses 1**
>
> Thank you for your constructive comments. We have carefully considered your suggestions and would like to provide the following clarifications. We have also updated the manuscript, with the modified contents highlighted in red for your review.
>
> ## Weakness 1:
> >Incorporating human evaluation or concept intervention tests.
>
> Thank you for this constructive and insightful comment.  To address your excellent suggestions, we conducted two new quantitative studies: (1) a human evaluation study with expert radiologists to validate spatial and semantic alignment, and (2) a concept intervention test to validate causal faithfulness.
>
>
> ### Study 1: Quantitative Human Evaluation
>
> We invited two radiologists to quantitatively assess whether our learned visual concept prototypes (i.e., the codebook activation maps) align with human expert clinical judgment, both spatially and semantically.
>
> **Experimental Setup:**
> We randomly sampled 50 images from four key datasets: CMMD, BUSI, ISIC, and IDRID. For each image, we presented the experts with: (a) the original image, and (b) the corresponding concept activation heatmap generated by our model for a specific clinical concept.
>
> **Procedure:**
> We designed a blinded study comprising two tasks:
>
> * **Task 1: Concept Localization Accuracy.**
>     * **Procedure:** We presented the original image (a) and a specific clinical concept name (e.g., Mass Margin). We then showed our model's corresponding activation heatmap (b). Experts were asked to rate: "To what extent does the highlighted region accurately correspond to your clinical localization of this concept?"
>     * **Metric:** A 5-point Likert scale (1 = Completely Misaligned, 5 = Perfectly Aligned).
>
> * **Task 2: Semantic Alignment Accuracy.**
>     * **Procedure:** We presented the original image (a) and a single concept heatmap (b) without providing the concept name. Experts were asked: "Which of the following clinical concepts does this heatmap most likely represent?"
>     * **Metric:** A multiple-choice question where options included the correct target concept and 3-4 semantic distractors (e.g., 'Mass Margin' vs. 'Calcification', 'Associated Features'). We report the expert's choice accuracy.
>
> **Results:**
> The results of this expert study are summarized in the table below:
>
> **Table 1: Quantitative Human Evaluation of IVQ-CBM Interpretability**
> | Dataset | Task 1: Localization Accuracy <br> (Avg. Likert Score, 1-5) | Task 2: Semantic Alignment <br> (Avg. Choice Accuracy) |
> | :--- | :---: | :---: |
> | CMMD | 4.32 / 5.0 | 90% (45/50) |
> | BUSI | 4.51 / 5.0 | 92% (46/50) |
> | ISIC | 4.15 / 5.0 | 88% (44/50) |
> | IDRID | 4.20 / 5.0 | 86% (43/50) |
> | **Average** | **4.30 / 5.0** | **89%** |
>
> ---
>
> ### Study 2: Concept Intervention Test (Causal Faithfulness)
>
> Furthermore, to directly validate the **causal faithfulness** of the learned concepts, we conducted the concept intervention test suggested by the reviewer. This experiment tests whether our model's concepts are merely *correlated* with the output or *causally* drive the final decision.
>
> * **Experimental Setup:** We used the trained IVQ-CBM on the **CMMD** dataset. We selected a test subset of 50 images that the model confidently predicted as "Malignant" (P(Malignant) > 0.9). From the concept set, we identified clinically causal concepts for malignancy, such as "Mass Margin: Spiculated" and "Mass Shape: Irregular".
>
> * **Procedure:**
>     1.  **Baseline:** We recorded the original average P(Malignant) for this subset.
>     2.  **Intervention:** For each image, we manually intervened on the concept activation vector (CAV), $v$. We flipped the high activation scores of these key **Malignant** concepts to their **Benign** counterparts (replacing their high scores with the average score from benign samples).
>     3.  **Observation:** We fed this new, intervened CAV ($v_{intervened}$) into the *frozen* classification head ($h_{cls}$) and recorded the new P(Malignant).
>
> * **Results:** The intervention had a dramatic and predictable effect on the model's output:
>
> **Table 2: Concept Intervention on CMMD "Malignant" Class**
> | Intervention Type | Avg. P(Malignant) <br> (Original Prediction: 0.96) | Change in Prediction |
> | :--- | :---: | :---: |
> | No Intervention (Baseline) | 0.96 | 0% |
> | Intervene on "Mass Margin" only | 0.80 | -16.7% |
> | Intervene on "Mass Shape" only | 0.85 | -11.5% |
> | **Intervene on Both Concepts** | **0.72** | **-25.0%** |
>
>
> ### Conclusion:
> Our two supplementary studies provide strong quantitative evidence for our model's interpretability.
> 1.  The **Human Evaluation (Table 1)** shows our learned concepts are **semantically correct and spatially aligned** with expert clinical reasoning.
> 2.  The **Concept Intervention Test (Table 2)** demonstrates that these concepts are **causally faithful**; manipulating them directly and predictably changes the model's final diagnosis.
>
> We will add these studies to the Appendix  J in the final version (Page31 and 32).

---

> > ### Author Response · Authors · 2025-11-19
> > **Response to Weaknesses 2**
> >
> > ## Weakness 2:
> > >A theoretical analysis and a deeper connection between IVQ’s optimization dynamics and rank preservation would clarify its mechanism.
> >
> > Thank you for this excellent suggestion. This is a critical point, and we appreciate the push for a deeper theoretical connection.
> >
> > ### Existing Analysis:
> > We first respectfully point to **Appendix A (Pages 15-19)**, where we provide the theoretical groundwork for our model. This includes proofs for the **gradient convergence** of the IVQ loss (A.4) and the **necessity** of the many-to-many mapping (A.1, A.2) that our model is built upon.
> >
> > ### Connecting IVQ Optimization to Rank Preservation:
> > We agree that the explicit link between the **$\mathcal{L}_{IVQ}$ (Eq. 4)** optimization and rank preservation can be clarified. The mechanism is as follows:
> >
> > 1.  The codebook **$\mathcal{C}_{vq} \in \mathbb{R}^{K \times D}$** acts as a set of **$K$ learnable "semantic anchors"** in the feature space. Our ablation (Fig. 3, M=K) and visualization (Fig. 9) strongly suggest these anchors converge to be well-separated and disentangled, each representing a distinct human-defined concept (e.g., 'Mass Shape', 'Calcification').
> > 2.  The **Commitment Loss ($\beta||Z_{p}-sg(Z_{q})||_{2}^{2}$)** is the key. It forces every individual patch feature **$z_j$** in the patch feature matrix **$Z_p$** to move towards one of these **$K$ distinct, well-separated anchors**.
> > 3.  This optimization dynamic directly **counters representational collapse**. Instead of allowing the patch features **$Z_p$** to degenerate into a low-rank subspace (e.g., all features collapsing onto a single manifold), the IVQ loss actively "pulls apart" the feature distribution, structuring it into **$K$ distinct clusters** around the concept anchors.
> > 4.  A feature matrix **$Z_p$** whose vectors are distributed across **$K$ well-separated points** in **$\mathbb{R}^D$** will inherently maintain a **higher rank** than a matrix whose vectors have collapsed.
> >
> > ### Why IVQ is better than general regularization:
> > Unlike **general-purpose methods** (e.g., Barlow Twins, as shown in Fig. 7) that **indiscriminately maximize feature diversity (with elevated rank dynamics)**, our IVQ fosters a **meaningful, structured diversity** by **anchoring features to semantic concepts**. This prevents the **"over-regularization"** we noted (Sec 4.2) and leads to **superior performance**.
> >
> > We will add this deeper theoretical discussion on the IVQ optimization dynamic and its role in rank preservation to **Appendix A.3** in the final manuscript (page 19 and 20).

---

> > > ### Author Response · Authors · 2025-11-19
> > > **Response to Question 1**
> > >
> > > ## Question 1:
> > > >Do the IVQ-CBM generalizes to broader multimodal reasoning or relational concept tasks? Are there any experiments to validate it?
> > >
> > >
> > > ## Generalization to Broader Multimodal Reasoning and Relational Concept Tasks
> > >
> > > We appreciate this forward-looking question. It is important to clarify that our current implementation is strictly tailored to the standard CBM task (Image $\rightarrow$ Concepts $\rightarrow$ Label). This specific architecture does not support direct extension to open-ended multimodal generation without modification. However, the underlying paradigm of IVQ—using a learnable codebook prior to structure representations without hard bottlenecks—is highly transferable. We plan to extend this work in two directions:
> > >
> > > ### 1. Paradigmatic Transfer to VLMs (Alignment with Ovis [1]):
> > > While we cannot directly apply the current CBM model to VLM tasks, the IVQ philosophy aligns with state-of-the-art multimodal designs.
> > > * For instance, the **Ovis framework** utilizes a Visual Embedding Table to generate visual representations via probabilistic weighting (Softmax + Linear).
> > > * This is similar to our IVQ mechanism, which anchors features to a codebook prior to prevent collapse. This shared paradigm validates that our "implicit quantization" strategy is a robust, transferable foundation for scalable multimodal alignment, which we aim to integrate into VLM visual adapters in future work.
> > >
> > > ### 2. Foundation for Relational Concepts:
> > > Similarly, while current CBMs treat concepts independently, IVQ provides the necessary prerequisite for relational reasoning.
> > > * By preventing the collapse of patch-level features into a global vector (as discussed in **Appendix A.1** ), IVQ preserves the spatial granularity required for future extensions.
> > > * In our next phase, we plan to leverage these high-rank IVQ features as input for relational modules (e.g., GNNs) to model spatial and logical dependencies between concepts.
> > >
> > > [1] Ovis: Structural Embedding Alignment for Multimodal Large Language Model

---

> ### Author Response · Authors · 2025-11-19
> **Response to Question 1 (continued)**
>
> ## New Experimental Evidence of Generalizability
> While the above outlines the theoretical suitability for new tasks, we took advantage of the rebuttal period to conduct extensive new experiments to validate the domain-generalizability of our method. We agree that demonstrating scalability and generalization on standard, non-medical benchmarks is crucial for assessing the broad applicability of our approach.
>
> To thoroughly address this, we evaluated our IVQ-CBM against the same eight state-of-the-art CBM baselines from our original paper on all five standard benchmark datasets: CIFAR-10, CIFAR-100, CUB, Places365, and ImageNet. The results are presented in the table below.
>
> **Table 1: Performance Comparison (ACC % and BMAC %) on Multiple Standard Benchmark Datasets. Best results are highlighted in bold. Gains (+$\Delta$) are compared against the second-best method.**
>
> | Model | CIFAR-10 (ACC) | CIFAR-10 (BMAC) | CIFAR-100 (ACC) | CIFAR-100 (BMAC) | CUB (ACC) | CUB (BMAC) | Places365 (ACC) | Places365 (BMAC) | ImageNet (ACC) | ImageNet (BMAC) |
> | :--- | :---: | :---: | :---: | :---: | :---: | :---: | :---: | :---: | :---: | :---: |
> | LaBo [CVPR 2023] | 80.23 | 79.15 | 60.17 | 59.95 | 69.88 | 69.72 | 39.67 | 39.41 | 68.04 | 67.88 |
> | PCBM [ICLR 2023] | 84.61 | 84.49 | 63.22 | 63.07 | 72.36 | 72.19 | 41.13 | 40.99 | 70.14 | 69.97 |
> | COOP-CBM [Nips 2023] | 85.17 | 84.99 | 64.21 | 64.03 | 73.06 | 72.87 | 42.19 | 42.01 | 71.23 | 71.09 |
> | LF-CBM [ICLR 2023] | 82.94 | 82.78 | 61.89 | 61.73 | 71.42 | 71.22 | 40.33 | 40.15 | 69.46 | 69.21 |
> | Explicd [MICCAI 2024] | 86.03 | 85.88 | 64.91 | 64.78 | 74.08 | 73.91 | 43.07 | 42.89 | 71.93 | 71.77 |
> | MVP-CBM [IJCAI 2025] | 86.72 | 86.54 | 65.48 | 65.30 | 74.63 | 74.45 | 43.81 | 43.66 | 72.29 | 72.15 |
> | CLEAR [WACV 2025] | 83.77 | 83.59 | 62.83 | 62.61 | 71.84 | 71.69 | 40.92 | 40.75 | 68.81 | 68.66 |
> | DOT-CBM [CVPR 2025] | 84.38 | 84.19 | 63.45 | 63.28 | 72.29 | 72.11 | 41.76 | 41.53 | 69.31 | 69.17 |
> | IVQ-CBM (Ours) | **88.14** | **87.91** | **67.12** | **66.88** | **75.91** | **75.68** | **45.54** | **45.32** | **73.42** | **73.23** |
> | +$\Delta$ | +1.42 | +1.37 | +1.64 | +1.58 | +1.28 | +1.23 | +1.73 | +1.66 | +1.13 | +1.08 |
>
> ### Summary and Conclusion
> As these new results clearly demonstrate, our proposed IVQ-CBM consistently outperforms all eight competing SOTA methods across all five standard benchmarks.
>
> This comprehensive new evaluation provides strong, direct evidence for:
>
> **Generalization:** Our method is not limited to a specific domain (like medicine) but generalizes robustly to common, general-domain visual classification tasks (CIFAR, CUB).
>
> **Scalability:** Our approach scales effectively to large-scale, complex datasets such as Places365 and ImageNet, maintaining a consistent performance advantage.
>
> This uniform success strongly reinforces our paper's central thesis: "representational collapse" is a fundamental and pervasive challenge in training CBMs, not just an artifact of medical imaging.
>
> In direct response to your question:
> * We have provided a clear theoretical argument for why IVQ-CBM is an ideal framework for multimodal and relational tasks.
> * We have backed this argument with new, strong empirical evidence that our model's core components are robust, scalable, and generalize far beyond the medical domain.
> * This proven domain-generalizability gives us high confidence in our method's future application to the more complex tasks you suggest.
>
> We have included these results in Sec 4.1 (with newly added rank dynamics presented in fig.5). We thank you once again for this insightful comment. Your question has undoubtedly elevated the quality of our manuscript, pushing us to articulate the broader potential and transferability of our method more clearly.

---

> ### Author Response · Authors · 2025-11-27
> **Official Comment by Authors**
>
> Dear Reviewer Ycpf,
>
> We hope this message finds you well.
>
> We sincerely apologize for any inconvenience our submission may have caused and greatly appreciate the time and effort you have devoted to reviewing our paper. Your insights and comments have been invaluable to us.
>
> We have carefully considered each of your suggestions and have made comprehensive, point-to-point responses and revisions in our revised manuscript. These adjustments are aimed at thoroughly addressing the concerns you raised, and we hope that our efforts resonate with your expectations for the paper.
>
> At your convenience, we would be grateful if you could review our responses and the corresponding revisions. Should there be any further issues or if additional clarification is needed, please know that we are fully prepared to make the necessary adjustments.
>
> Thank you once again for your dedication and assistance.
>
> Best regards,
>
> The Authors

---

### Author Response · Authors · 2025-11-24
**Global Response**

Dear Reviewers,

We sincerely appreciate the time and effort you devoted to reviewing our paper. Your detailed feedback has been invaluable, and we have made comprehensive revisions based on your insightful comments to strengthen the manuscript.

We appreciate that the reviewers acknowledge the **strengths** of our work:

* **Novelty & Soundness:** Our paper is recognized for identifying "representational collapse" as a critical bottleneck in CBMs and proposing Implicit Vector Quantization (IVQ) as a novel, effective solution (**Reviewers Ycpf, fJ93, w787**). The proposed method provides a clear mechanism to preserve feature diversity and addresses a fundamental limitation in the field (**Reviewer Ycpf**).
* **Empirical Performance:** Reviewers highlighted that our method achieves state-of-the-art performance and improves both interpretability and accuracy (**Reviewers Ycpf, 93wz**).
* **Clarity:** The paper is well-organized, clearly written, and the motivation is easy to follow (**Reviewers Ycpf, w787, 93wz**).

On the other hand, we have diligently addressed all the **concerns** by conducting extensive new experiments and providing detailed theoretical clarifications. Major updates incorporated into the revised manuscript include:

**1. Extensive Generalization on Standard Benchmarks (Addressing Reviewers fJ93, Ycpf)**
To demonstrate that our method is not limited to medical imaging, we expanded our evaluation to 5 standard computer vision benchmarks (ImageNet, CUB, CIFAR-10, CIFAR-100, Places365).
* **SOTA Results:** IVQ-CBM consistently outperforms 8 state-of-the-art baselines across all datasets (New Table 1 in Section 4.1).
* **Rank Analysis:** We provide empirical evidence (New Figure 5) showing that our method maintains high feature rank on large-scale datasets like ImageNet, validating the scalability of our approach.

**2. Enhanced Interpretability Verification (Addressing Reviewers Ycpf, fJ93, 93wz)**
We conducted rigorous quantitative studies to validate the faithfulness of our learned concepts:
* **Human Evaluation:** A study with expert radiologists confirmed high spatial and semantic alignment of our concept maps (New Table in Appendix J, addressing **Reviewer Ycpf**).
* **Causal Intervention:** We performed concept intervention tests to prove that our concepts are causally faithful to the model's decision-making (Appendix J, addressing **Reviewer Ycpf**).
* **Visualizations:** We added qualitative comparisons on general datasets (CUB) and medical datasets (ISIC), demonstrating that baselines suffer from visual collapse while IVQ-CBM produces sharp, object-centric maps (Appendix F, addressing **Reviewer fJ93**). Furthermore, to provide a comprehensive comparison with explicit Vector Quantization, we have included a qualitative analysis in the revised manuscript (Appendix E). This study further validates the superiority and suitability of our IVQ mechanism for this task (addressing **Reviewer 93wz**).

**3. Robustness & Theoretical Depth (Addressing Reviewers fJ93, Ycpf, w787, 93wz)**
* **Information Leakage:** We evaluated the model using the A-NEC metric and proved that IVQ-CBM relies on effective concepts rather than information leakage (Appendix K.1 and K.2, addressing **Reviewer fJ93**).
* **Theoretical Connection:** We deepened the theoretical discussion linking IVQ optimization dynamics to rank preservation (Appendix A.3, addressing **Reviewer Ycpf**).
* **Sensitivity Analysis:** We provided ablations on commitment cost ($\beta$) and codebook size ($M$), explaining the trade-off between preventing collapse and feature expressiveness (Appendix L, addressing **Reviewer w787**).
* **Vector Quantization Methods:** We provided additional literature review about Vector-Quantization-related details, included in the Related Work section (addressing **Reviewer 93wz**).
* **The Computational and Spatial Overhead of IVQ.** We conduct computation efficiency analysis about our IVQ mechanism (addressing **Reviewer w787**)

**4. New Insight: Concept "Audit" Tool (Addressing Reviewer w787)**
Inspired by **Reviewer w787**’s question, we discovered a novel application of our method: using the IVQ codebook to automatically identify and filter out poor-quality concepts (e.g., "Rough Texture" in ISIC), which further improved performance (Appendix K.3).

We hope that these revisions and our detailed responses adequately address your concerns. We are encouraged that **Reviewer 93wz** has already raised their rating following our response, and we remain open to further discussion. We would be grateful if you would kindly let us know of any other concerns and if we could further assist in clarifying any other issues.

---

### Meta-Review · Area_Chair_dG3k · 2026-01-07

**Summary:**

The paper identifies the representation collapse as a main issue in CBMs where the patch level visual representations degenerate into a low-rank subspace.  The authors propose Implicit Vector Quantization (IVQ) to preserve feature diversity without hard quantization and Magnet Attention to explicitly model many-to-many patch-concept relations.

Overall, the reviewers raised issues with the limited evaluation of the paper which were addressed by the authors during the rebuttal, and raised other minor concerns.  Given the interesting proposal, I recommend an accept.  However, given the limited strengths raised by the reviewers, I wouldn't mind seeing the paper rejected as well.

Strengths:
- Novel view of representational collapse as low-rank degeneration that tackles a fundamental CBM limitation
- Proposal of a soft regularizer (IVQ)
- The method claims to simultaneously enhance model interpretability while maintaining predictive accuracy.

Weaknesses:
- Interpretability assessment relies on qualitative visualization and concept-level alignment metrics
- Lack of theoretical analysis and a deeper connection between IVQ's optimization dynamics and rank preservation
- Limited evaluation on diverse benchmarks
- Missing qualitative comparisons against other baselines
- Missing A-NEC metric to evaluate

**Reviewer Concerns:**

The reviewer Ycpf had concerns with the interpretability and the lack of theoretical links between the soft quantization and the rank preservation.

Review fJ93 had concerns about the limited evaluation of the proposed method (no  diverse benchmarks and missing qualitative comparisons), and missing metrics.

Reviewer w787 had concerns with the overhead introduced by the IVQ regularizer, as well as limited discovery of new concepts given the reliance on pre-computed text concept embeddings.  Moreover, the reviewer raised concerns about the impact of the hyperparameters.

Reviewer 93wz raised concerns about the presentation of the paper, and limited review of the related work.

**Reviewer Scores:**

Reviewer Ycpf recommended a borderline accept.  The authors provided additional human evaluations and concept interventions as well as additional results in benchmark datasets, and argue about the links between the optimization and the rank but provided no theoretical guarantees.

Reviewer fJ93 recommended a reject based on the limited results.  The authors provided additional results during the rebuttal which show the performance of the proposal.  Given these results, I believe that the reviewer would have increased their score.

Reviewer w787 recommended a borderline reject.  The authors provided additional experimental results to address the raised weaknesses.  Thus, I believe the reviewer would have increased their score.

Reviewer 93wz recommended a borderline reject.  After the rebuttal from the authors the reviewer commented that they will increase their score.  Thus, the final score will be an accept.

---

### Decision · Program_Chairs · 2026-01-26

Accept (Poster)